# Prediction and validation of protein intermediate states from structurally rich ensembles and coarse-grained simulations

Laura Orellana[1], Ozge Yoluk[1], Oliver Carrillo[2], Modesto Orozco[3] & Erik Lindahl[1]

Protein conformational changes are at the heart of cell functions, from signalling to ion transport. However, the transient nature of the intermediates along transition pathways hampers their experimental detection, making the underlying mechanisms elusive. Here we retrieve dynamic information on the actual transition routes from principal component analysis (PCA) of structurally-rich ensembles and, in combination with coarse-grained simulations, explore the conformational landscapes of five well-studied proteins. Modelling them as elastic networks in a hybrid elastic-network Brownian dynamics simulation (eBDIMS), we generate trajectories connecting stable end-states that spontaneously sample the crystallographic motions, predicting the structures of known intermediates along the paths. We also show that the explored non-linear routes can delimit the lowest energy passages between end-states sampled by atomistic molecular dynamics. The integrative methodology presented here provides a powerful framework to extract and expand dynamic pathway information from the Protein Data Bank, as well as to validate sampling methods in general.

[1] Department of Theoretical Physics, KTH Royal Institute of Technology, Box 1031, 171 21 Solna, Sweden. [2] Scuola Normale Superiore de Pisa, Dipartimento di Fisica, Piazza dei Cavalieri, 7, 56126 Pisa, Italy. [3] Joint BSC-IRB Research Program in Computational Biology, Institute for Research in Biomedicine (IRB) Barcelona, C/Baldiri Reixac 10, 08028 Barcelona, Spain. Correspondence and requests for materials should be addressed to L.O. (email: laura.orellana@scilifelab.se).

Proteins function as sensors that cycle between different states in response to external stimuli. In general, stable conformers captured experimentally represent the end states of the functional cycle, while short-lived or highly flexible intermediates along the transition—which often hold the key to understand molecular mechanisms—are difficult to trap. Although a host of theoretical strategies have been developed to sample transition pathways, the intrinsic difficulty to predict the routes for conformational change and the lack of experimentally resolved intermediates hamper the validation of path-sampling methods.

Hitherto, in silico pathways are typically evaluated on the basis of stereochemical quality of the structures or by tracking progression along system-defined coordinates[1,2]. However, the selection of heuristic collective variables (CVs) is non-trivial and dimensionality reduction can be problematic[3]. Structural quality or progression along a few order parameters does not assure that a pathway samples biologically relevant routes to connect end-states. An interesting approach, proposed by Weiss and Levitt[4], is to benchmark path-sampling methods against proteins with at least three distinct states solved, and measure how close the sampled pathway spontaneously approaches known intermediates in terms of root mean square deviation (rMSD). Still, such procedure cannot assess the feasibility of the movements or to what extent they correspond to the biological motions. To address this issue we propose to take a step beyond simple two- or three-state benchmarking by making an ensemble-level analysis that considers all structural information available in the Protein Data Bank (PDB) for a given protein. Although there have been works systematizing protein motions in databases[5], a general and reliable framework to unlock and expand the pathway information contained in structural ensembles is still missing.

Principal component analysis (PCA)[6] is a powerful technique to decode ensemble motions and has been successfully applied to extract principal components (PCs) from experimental ensembles and to evaluate normal modes (NMs)[7–10], as well as essential motions obtained from molecular dynamics (MD) simulations[11]. For example, McCammon and co-workers[12–14] showed the utility of PCs obtained from X-ray structural ensembles as CVs to track MD; a recent work used PCs to estimate free-energies of transitions[15]. Here we build on the idea to use the two dominant PCs as complex multidimensional reaction coordinates to reveal the direction of ensemble-encoded conformational changes. The key to our analysis is a selection criterion different from previous ensemble-based studies[16], more focused on the quantity rather than the quality of the sampling by experimental structures. We argue that, only when the solved structures (regardless of their number) sample at least three different interconnected conformations, the PCs provide optimal CVs to highlight transition paths in the conformational landscape. By focusing on five structurally rich and diverse model systems we demonstrate that X-ray ensemble PCA accurately clusters resolved structures into different functional states. We show that for these proteins, the representation of the conformational space is robust even with minimal numbers of structures as long as they are well distributed along interconnecting paths. The projection of experimental conformers onto the PC-space provides an excellent visual representation of the structural landscape for a protein with known intermediates. Importantly, it also allows for immediate evaluation of the sampled pathways as it gives information on the natural sequence of on-pathway intermediates, which in turn can reveal information on their functional significance.

On the basis of the proposed ensemble PCA, we compare the performance of a novel coarse-grained (CG) path sampling algorithm named eBDIMS, using elastic network model (ENM) driven[10] Brownian[17] simulations, with several well-established methods as well as with state-of-the-art MD simulations (see Methods). Path sampling algorithms span from simple morphings[18] based on interpolations in Cartesian[19] or internal coordinates[20], to geometrical targeting[21] or atomistic techniques based on energy minimization[4]. A number of MD-based approaches are applied to explore transitions, for example the nudged elastic band[22] or the 'strings' method[23,24], as well as enhanced sampling algorithms such as conformational flooding[25], metadynamics[26], accelerated MD[27] or the accelerated weight histogram (AWH) method[28]. Although accurate, these techniques are computationally expensive and limited to small systems and short timescale transitions. CG-models[29,30], where each residue is reduced to a few beads interacting by simple potentials, minimize computational costs. Among CG-methods, ENMs[31,32] are conceptually simple but capable of predicting accurately conformational changes[33]. Despite reducing protein architecture to a minimalist network of Cα-carbons connected by springs, NMs computed from the ENM potential describe transitions between X-ray pairs with surprising precision[5,34–36] and reproduce the flexibility from experimental ensembles or long MD simulations[8–10,37,38]. For years, ENMs have been at the core of transition methods from simple interpolations[39], to two-state ENMs[2,40]. Being limited to an equilibrium basin, pathway generation requires iterative computation and deformation along selected NMs, which can produce stereochemical distortions. Although these issues can be reduced applying internal coordinates[20], structure corrections[41], or just using the modes to bias more realistic simulations[42,43], mode selection still poses a problem. Here the use of the network potential in the context of a BD simulation avoids unrealistic structure deformations and provides spontaneous sampling along the relevant modes. We show that, compared with other approaches, eBDIMS smoothly samples the experimentally encoded motions, and can predict the sequence of intermediates as accurately as Climber, an atomistic method[4] based on the Energy Calculation and Dynamics (ENCAD) molecular-mechanics force-field, but with the versatility of a simulation.

The integrated analysis of the PCs and the in silico pathways provides novel insights into the conformational changes of the studied proteins. We further demonstrate that simple algorithms such as eBDIMS or Climber accurately sample the conformational space given by experimental data and MD, predicting the lowest energy paths defined by transition intermediates. In conclusion, the methodology outlined here provides a powerful framework to extract and expand dynamic information from the rapidly growing PDB to evaluate sampling methods or even the functional status of new experimental structures.

## Results

**Pathway validation by PCA of structurally-rich ensembles.** We studied five proteins of different size, stoichiometry and motions (Supplementary Table 1) that specifically have well-defined intermediates between end-states (either described in the literature or in the PC1-2 space; see Methods), to perform robust ensemble PCA (Fig. 1, Supplementary Fig. 1, Supplementary Table 2) for eBDIMS pathway validation (see further examples in Supplementary Fig. 2 and Supplementary Table 3). The trajectories were compared with linear (NOMAD-Ref[44], MinActPath[45] and iENM[46]) and non-linear path-sampling algorithms (iMods[20], NMSIMs[47] and Climber[4]) (Supplementary Tables 4–5). We also created ensembles along the lowest frequency NMs[10] to study intrinsic motions. The apo-like, resting or inactive form was the reference

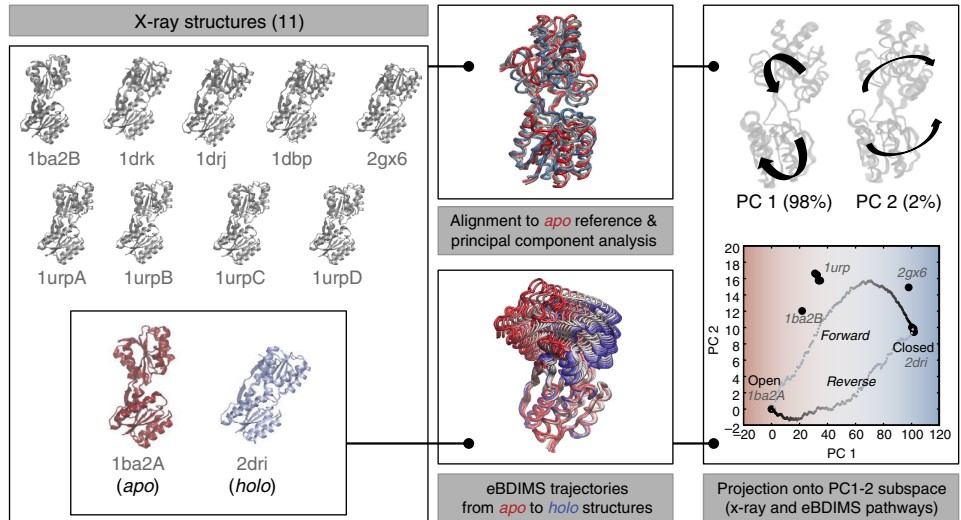

**Figure 1 | Reconstruction of the structural landscape by PCA from structurally rich ensembles and generation of transition pathways by eBDIMS.** The workflow is shown for a typical example (RBP, details in Fig. 2): structures are collected from the PDB (left panel) and the end-states are selected based on the literature; the ensemble is aligned to the inactive or apo state of the protein for PCA, and forward and reverse trajectories are independently computed with the eBDIMS sampling method (middle panel). Finally, the two principal PCs are used for projection analysis of the trajectories (right panel), allowing for straightforward detection of on-pathway intermediate states.

for PCA; all analyses focused on the two major PCs, covering >70% of the variance (Supplementary Table 2) and 95% of the transitions (Supplementary Table 6). For non-linear methods, trajectories were computed in forward/reverse directions, and eBDIMS pathway asymmetry quantified (see Methods). Finally, eBDIMS trajectories were compared with the free energy landscapes (FELs) sampled by MD for three representative examples. Interactive PC1-2 plots are provided in Supplementary Data 1. An alternative version can be found at http://data.tcblab.org/doi/10.1038/ncomms12575/Transitions.html.

***Escherichia coli* ribose-binding protein (RBP).** RBP is a periplasmic protein that binds ribose with a 6Å hinge motion of two similar domains. Although its crystallographic ensemble only contains eleven structures, they cover the entire closing process; all the conformers exist at equilibrium and ribose concentration shifts their distribution[48]. We selected the open (free) structure 1BA2 and the closed (bound) 2DRI as the end-states (Fig. 2a). The first PC (PC1), which describes domain closing (bending), accounts for most of the variance (97%) of the transition, while the second (PC2) describes a subtle oscillation (twisting; 2%) (Fig. 2b, Supplementary Tables 2 and 6). PC1 broadly separates the ensemble into three clusters of decreasing opening angle (Supplementary Fig. 3a): first, the unbound conformers, then intermediates 1URP and 2GX6, and finally the ligand-bound cluster. There is an excellent alignment of the first NM with the distribution of experimental structures along the path; however, this mode is not the best aligned with the difference transition vector between the end points (Supplementary Fig. 4a and Supplementary Table 7). Trajectory projection onto the PC1-2 subspace shows that the examined methods differ notably in how they sample the X-ray motions. Although they all approach the existent intermediates (Fig. 2c, right), reaching as close as 0.5–1.5 Å rMSD (Fig. 2d and Supplementary Table 5), the PC1-2 Euclidean distances discriminate between paths that resemble a straight interpolation and the ones that explore the subtle PC2 oscillations that accompany protein closure. Interestingly, eBDIMS and Climber converge in the reverse pathway through a straight-like route with a slightly smaller

rotational deviation along PC2. However, the asymmetry score is low (0.1; Supplementary Fig. 5), and as shown by MD, both routes are actually sampling the edges of the same low energy passage connecting the end-conformations (see Supplementary Fig. 6a and FEL below).

***E. coli* 5′-nucleotidase (5′-NTase).** 5′-NTase, an enzyme that hydrolyses nucleotides, is formed by two globular domains linked by an α-helix. Upon binding, an unusual 96° ball-and-socket rotation (rMSD 9.3 Å) moves the ligand along the interdomain surface into the catalytic site. The X-ray ensemble (sixteen structures) covers domain closure, with intermediates trapped by Cys-bridges. We selected the open 1OID and closed (RNA-bound) 1HPU structures as end-states (Fig. 3a). Again, PCA decomposes the ensemble into a major PC1 capturing most of the ball-and-socket motion (95%) of the transition (Supplementary Fig. 3b) and a minor PC2 tracking a subtle orthogonal rotation (4%) (Fig. 3b and Supplementary Tables 2 and 6). PC1 alone clusters the structures into three functional groups: the open structures (1OID and others), the intermediates (1OI8 and 4WWL), and the catalytically competent closed state (1HPU, 1HO5), while PC2 further helps to rank path sampling algorithms (Fig. 3c). The intermediates are sequentially visited by all methods reaching as close as 1.8 Å rMSD with eBDIMS (Fig. 3d and Supplementary Table 5), but projection onto PC2 reveals instabilities in some of the ENM algorithms. Once again, the NMs of the end-states perfectly align with the distribution of experimental structures (not shown) and the direction of conformational change (Supplementary Table 7). Both eBDIMS and Climber smoothly sample the forward transition, which departs along PC2, while the reverse paths proceed straightforward. As in RBP, pathway asymmetry is low (0.2; Supplementary Fig. 5), indicating that both routes in fact explore the same low energy through between the end-states, a notion further supported by spontaneous sampling from unbound forms in MD (Supplementary Fig. 6b, see below).

***Aquifex aeolicus* ribonuclease III (RNaseIII).** RNaseIII is an $Mg^{2+}$-dependent enzyme that modulates gene expression by

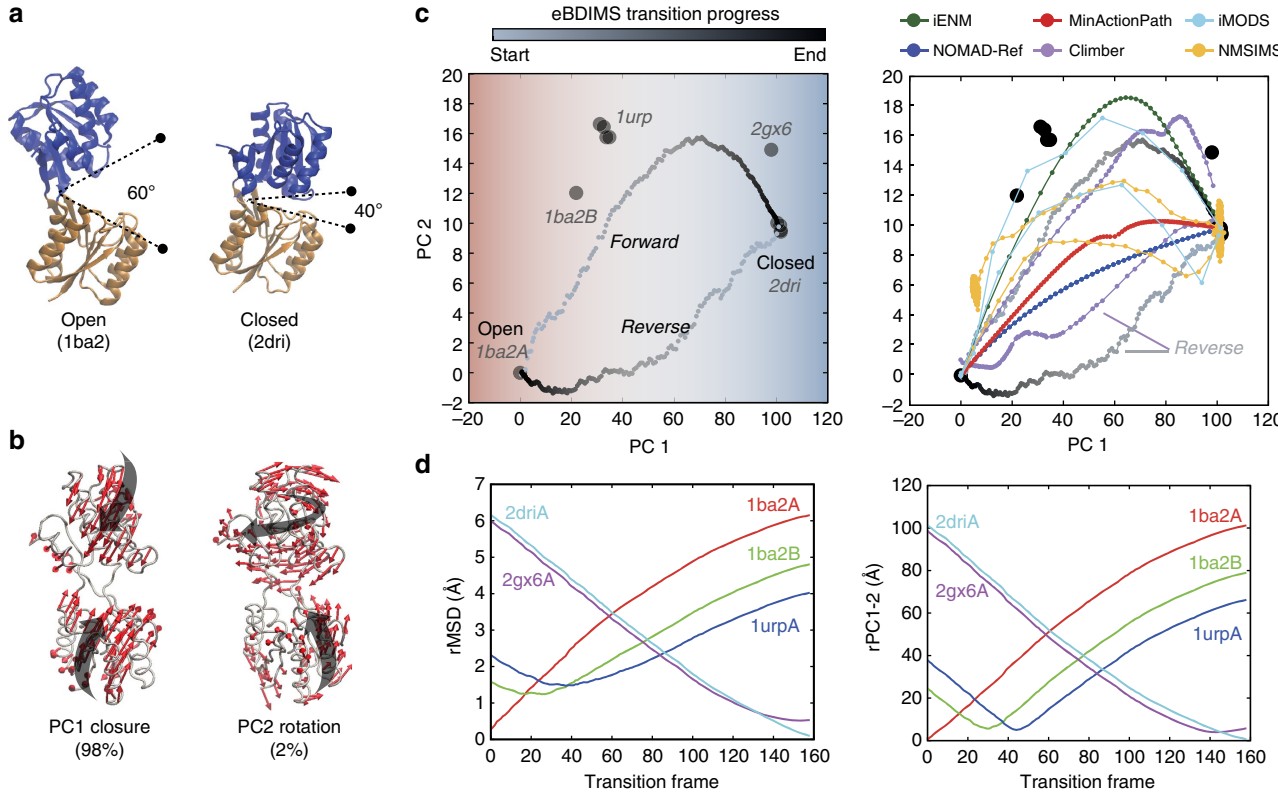

**Figure 2 | Conformational transition of *E. coli* ribose-binding protein (RBP).** (**a**) Crystallographic unbound (1ba2) and ribose-bound (2dri) state of RBP; the two ligand-binding domains are coloured in blue and orange. (**b**) Principal components of the X-ray ensemble (11 structures) track domain closing and subtle rotation versus the reference 1ba2. (**c**) Left: projection of the ensemble structures and the eBDIMS trajectories onto the PC1-2 subspace; note how PC1 separates the crystal structures into three–four clusters (shown as red (unbound), white (intermediates) and blue (bound) regions). Right: comparison between the forward pathways computed by eBDIMS, iENM, NOMAD-Ref, MinActionPath and Climber. Reverse pathways generated by eBDIMS, iMODS, NMSIM and Climber also shown. (**d**) rMSD and PC1-2 distance between the forward trajectory and the sequence of crystallographic intermediates.

cleaving double-stranded RNA (dsRNA). It is a symmetric homodimer where each subunit is composed of two domains (an RNaseIII domain, RIIID, and a dsRNA-binding domain, dsRBD) separated by a flexible linker. RIIID dimerization forms a 'catalytic valley' to accommodate dsRNA; the dsRBDs need to rotate dramatically (180°) to position RNA on it. However, this region is negatively charged, requiring $Mg^{2+}$-coordination to mitigate dsRNA repulsion. In absence of $Mg^{2+}$, dsRNA binds outside the valley in a 'non-catalytic' form (1YYO, 2NUE), but with $Mg^{2+}$ present, it moves inside the valley leading to a 'catalytic' form (2NUG, 2NUF, 2EZ6). Since a true apo structure is missing, the 11 RNA-bound crystallized structures represent intermediate snapshots in dsRNA processing between non-catalytic/catalytic states. Therefore, we focused on the transition between the non-catalytic complex RNaseIII–RNA4 (1YYO), and the pre-catalytic RNaseIII–RNA3 (1YYW), where dsRNA is reorienting (Fig. 4a). Between these two conformers (18 Å rMSD), there is a well-characterized intermediate, RNaseIII E110Q–RNA2 (1YZ9; ref. 49). Here the complexity of the movements decomposes the ensemble into two similarly weighted components (Fig. 4b and Supplementary Table 2): PC1 describes dsRBD-arms opening or 'breathing'[49] (51% variance), while PC2 tracks their concerted rotation (43% variance) (Supplementary Fig. 3c) capturing most of the transition (see Supplementary Table 6). Altogether they separate the structures into four functional groups: the 1YYO cluster, the intermediate 1YZ9, the pre-catalytic 1YYW cluster and catalytic 2EZ6 cluster (Fig. 4c, left). Along PC1 the structures separate into only three groups according to dsRBD-opening,

because the closed non-catalytic and catalytic complexes are not differentiated; PC2 clearly distinguishes their opposite orientations (with the RNA-binding surface looking outwards/inwards to the catalytic valley). Four of the methods (Fig. 4c, right) cannot track this challenging transition, which both eBDIMS and Climber sample smoothly visiting the 1YZ9 intermediate within 4 Å rMSD (Fig. 4d, Supplementary Table 5 and Fig. 6a,b). As above, the lowest NMs point to the nearest intermediate rather than to the transition direction (Supplementary Fig. 4b and Supplementary Table 7); at the 1YYW bifurcation, they split into two directions pointing back to 1YYO or to the $Mg^{2+}$-bound region (not shown). Here, the asymmetry score is high (0.4; Supplementary Fig. 5) but the trajectories approach on-pathway X-ray structures in both directions: the forward path crosses the 1YZ9 point as expected, and the reverse deviates along PC2 approaching the region populated by $Mg^{2+}$-bound structures. In fact, the transition from the pre-catalytic (1YYW) towards the catalytic state (2EZ6) is the next natural step along the RNaseIII cycle with $Mg^{2+}$ present. This suggests a multi-step mechanism (Fig. 6a,b) agreeing with experimental models[49] in which, as RNA binds (1YYO), the dsRBDs first separate along PC1 (crossing 1YZ9) and once they are wide-open, start to rotate along PC2 to reach 1YYW (90° rotation, forward). Then, as the bias (that is, $Mg^{2+}$ *in vivo*) favours closing again (reverse), the dsRBDs naturally approach the catalytically competent region (180°) up to a point where they start to rotate back to resting position (1YYO). Thus, the eBDIMS trajectory suggests that the topologically accessible route to relax and close the protein naturally approaches the catalytically

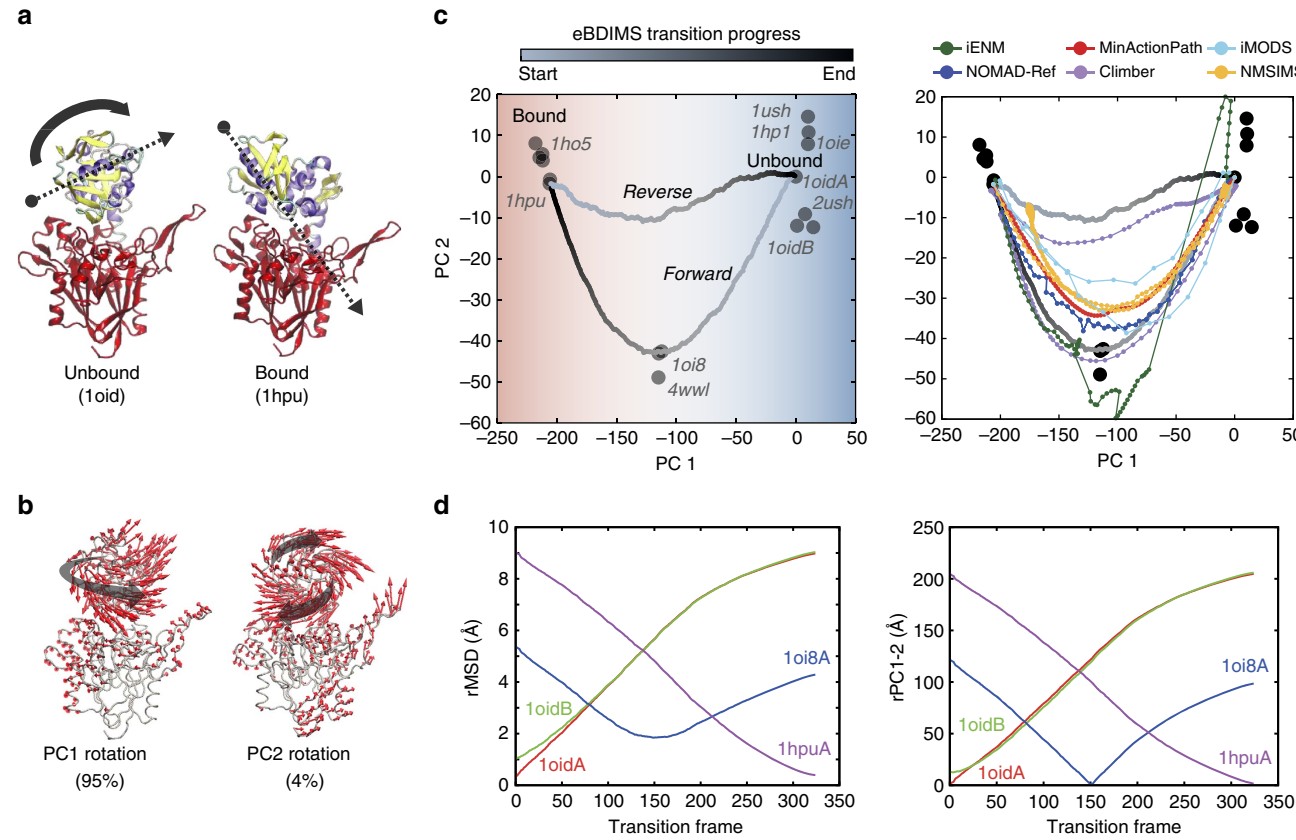

**Figure 3 | Conformational transition of *E. coli* 5′-nucleotidase (5′NTase).** (**a**) Crystallographic unbound (1oid) and nucleotide-bound (1hpu) states, showing one domain in red and the other coloured by secondary structure to visualize the ball-and-socket rotation (grey arrows). (**b**) Principal components of the X-ray ensemble (16 structures) decompose the complex rotation into a major and a minor rotation versus the reference 1oid. (**c**) Left: projections of the X-ray ensemble structures and the eBDIMS trajectories onto the PC1-2 subspace; PC1 alone separates the crystal structures into three clusters (same colour-coding as in Fig. 2). Right: comparison between the forward pathways computed by eBDIMS, iENM, NOMAD-Ref, MinActionPath and Climber. Reverse pathways generated by eBDIMS, iMODS, NMSIM and Climber also shown. (**d**) rMSD and PC1-2 distance between the forward trajectory and the sequence of crystallographic intermediates.

competent form. Our eBDIMS simulation does not mimic the presence of $Mg^{2+}$, but the fact that it spontaneously approaches the region populated by such conformations implies that the transition to a catalytic complex is pre-conditioned by the peculiar domain arrangement independent from electrostatic effects. The MD spontaneous sampling from the unbound structures seems to supports this, hinting at two possible different pathways (Supplementary Fig. 6c).

***Oryctolagus cuniculus* sarco-endoplasmic reticulum $Ca^{2+}$-ATPase (SERCA).** The SERCA pump is the best-studied P-type ATPase, which transport ions across cell membranes. There are >60 SERCA structures bound to ligands and ATP analogues, which cover nearly all the catalytic cycle. The cytosolic 'headpiece' domains, A (actuator), N (nucleotide binding) and P (phosphorylation) undergo translations and rotations as they bind and hydrolyse ATP. These motions are coupled to piston-like movements of transmembrane helices (TM) reshaping $Ca^{2+}$ accessibility. The pump cycles between E1/E2 conformations: in the E1 state (E1-free), the $Ca^{2+}$ high-affinity sites facing the cytoplasm are occupied (E1-2$Ca^{2+}$) favouring ATP binding (E1-2$Ca^{2+}$-ATP); then nucleotide hydrolysis triggers $Ca^{2+}$ transport (E2-2$Ca^{2+}$) releasing ions into the lumen (E2-free). We focused on the transition from the open E1-2$Ca^{2+}$ state (2C9M), with a splayed-headpiece, to the closed-headpiece

E1-2$Ca^{2+}$-ATP structure (1T5S) (rMSD 14 Å) (Fig. 5a), locked onto an ATP-binding pocket. As with RNaseIII, the complexity and amplitude of the motions along the catalytic cycle yields similarly weighted components (Fig. 5b and Supplementary Table 2): PC1 (57% variance), which tracks most of the E1->E2 ion pumping, and PC2 (28% variance), which describes 95% of the A/P closure to bind ATP (Supplementary Table 6); both PCs correlate with heuristic variables (Supplementary Fig. 3d). Altogether, these motions separate structures into seven clusters (Fig. 5c, left), with E1-2$Ca^{2+}$ showing great dispersion due to headpiece mobility.

For this transition, we were not aware of well-identified crystallographic intermediates. However, PCA neatly distributed structures into three groups along PC2: a cluster of E1-$Mg^{2+}$-bound structures (4H1W, 3W5A and 3W5B), visited by the eBDIMS closing trajectory (Fig. 5d, Supplementary Table 5 and Fig. 6c,d), appears between the most open E1-2Ca (2C9M and 1SU4) and most closed E1-2$Ca^{2+}$-ATP bound structures (1T5S cluster). Although most methods track this transition (Fig. 5c, right), some become unstable in the intermediate region, where the lowest NMs from nucleotide-bound/intermediate states split onto two orthogonal directions (Supplementary Fig. 4c, Supplementary Table 7).

To further test the versatility of eBDIMS, we reconstructed the structure 4NAB, a catalytically incompetent mutant E309Q. This structure was not included in the ensemble since the

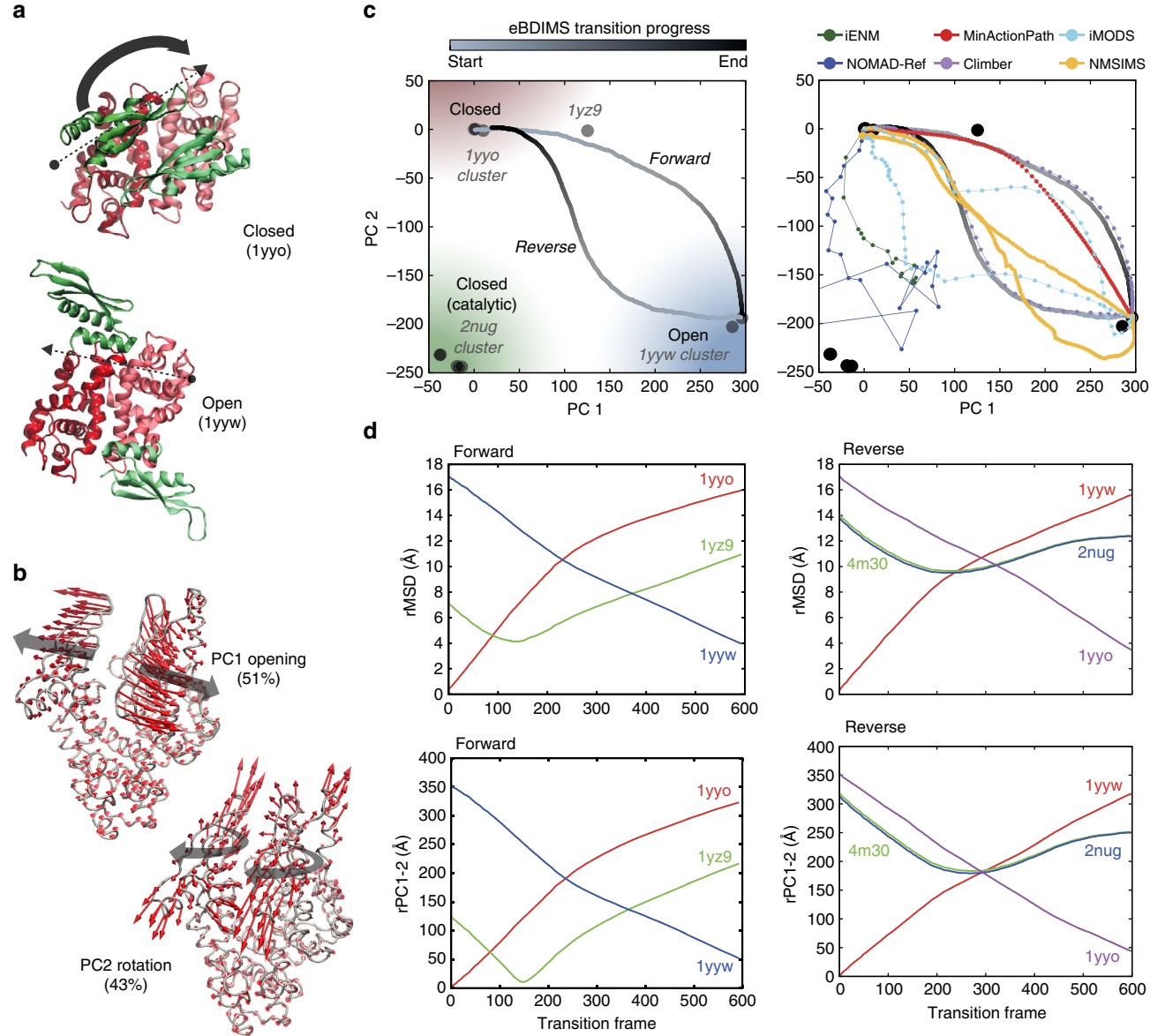

**Figure 4 | Conformational transition of *A. aeolicus* RNA endonuclease III (RNaseIII).** (**a**) Crystallographic dimers for closed non-catalytic state (1yyo) and open pre-catalytic state (1yyw), showing the dsRNA-binding domains (RBDs) in green and the rest of the protein in red surface; both structures are bound to dsRNA and represent different stages in re-orienting the substrate for catalysis. (**b**) Dominant PCs of the X-ray ensemble (11 structures) track RBD arms separation and rotation versus the reference 1yyo structure. (**c**) Left: projections of the ensemble and the eBDIMS trajectories onto the PC1-2 subspace reveal an hysteresis-like cycle with a tight sequence of concerted motions; PC1-2 separate the crystal structures into four clusters (shown as red (closed non-catalytic), white (intermediate), blue (open pre-catalytic) and green (closed catalytic) regions). Right: comparison between the forward pathways computed by eBDIMS, iENM, NOMAD-Ref, MinActionPath and Climber. Reverse pathways generated by eBDIMS, iMODS, NMSIM and Climber also shown. (**d**) rMSD and PC1-2 distance between the forward (left) and reverse (right) trajectories and the crystallographic intermediates approached.

A-domain is missing, but the loops connecting to TM1-2 allowed eBDIMS to approximate its position (Supplementary Fig. 7a). Upon projection, 4NAB surprisingly appeared as a potential topological intermediate in the reverse path (Fig. 5c left, red dot). Although here intermediates are present in forward/reverse routes, the subtle asymmetry (0.15; Supplementary Fig. 5) and similar sampling of heuristic variables (Supplementary Fig. 7b,c), suggests identical opening/closing routes. Inspection of the MD FEL supports this view, and hints again that non-linear pathways delimit the edges of a single low-energy trough with X-ray intermediates on both sides (see below).

***Gloeobacter violaceus* ligand-gated ion channel (GLIC).** Pentameric ligand-gated ion channels form a large family of membrane proteins with a central role in signal transduction, transmitting ligand binding through the opening of their ion-conducting pore. The proton-gated channel GLIC[50] has been intensely studied as a model for eukaryotic counterparts. Crystals of closed GLIC were recently determined[51], which together with locally closed[52] and open structures[53,54] track the gating mechanism. Upon $H^+$-binding, the extracellular domain undergoes a contraction (un-blooming) propagated to the intracellular region, triggering the tilting of pore-lining M2 helices in a cooperative iris-like motion that opens the gate; then

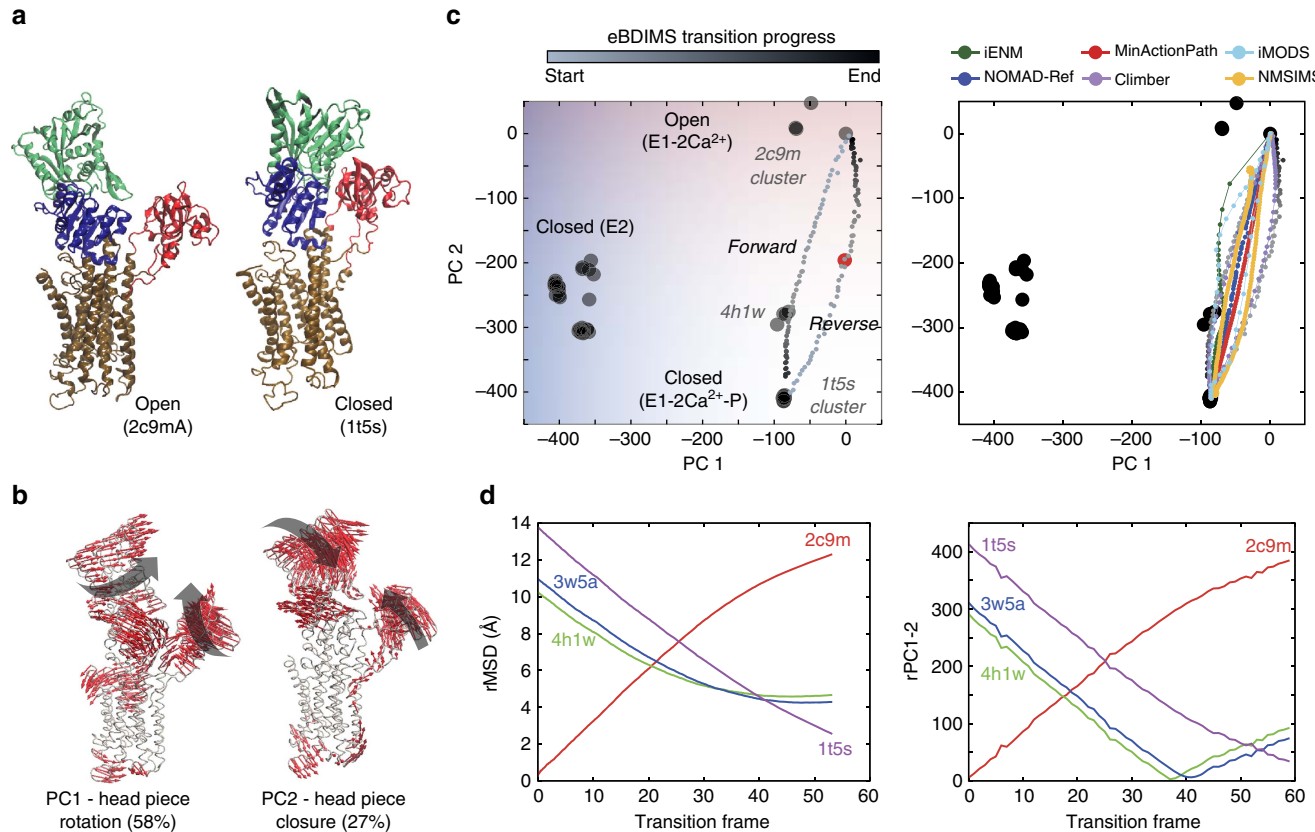

**Figure 5 | Conformational transition of *O. cuniculus* sarcoplasmic Ca$^{2+}$-pump (SERCA).** (**a**) Crystallographic E1-Ca$^{2+}$ bound open (2c9m) and E1-Ca$^{2+}$- nucleotide bound (1t5s) state, showing the headpiece domains coloured in red (A), blue (P) and green (N) and the TM helices in brown. (**b**) Principal components of the X-ray ensemble (65 structures) describe the E2->E1 pumping transition (PC1) and the closing of the headpiece (PC2). (**c**) Left: projections of the X-ray ensemble structures and the eBDIMS trajectories onto the PC1-2 subspace; PC2 separates the crystal structures into open (red) and closed (blue); the reconstructed structure 4nab is shown as a red dot. Right: comparison between the forward pathways computed by eBDIMS, iENM, NOMAD-Ref, MinActionPath and Climber. Reverse pathways generated by eBDIMS, iMODS, NMSIM and Climber also shown. (**d**) rMSD and PC1-2 distance between the forward trajectory and two crystallographic intermediates.

the quaternary twist of the subunits locks the receptor in the conducting state.

The GLIC X-ray ensemble contained 46 near-intact pentamers (Supplementary Table 2). We selected the structures 4NPQ and 4HFI as representatives of the resting and H$^+$-bound conducting states, respectively (Fig. 7a). In this case the transition is subtle (rMSD = 2.66 Å) but requires cooperative motion of five subunits. Here the main PCs are similarly weighted (Fig. 7b): while PC1 (42% variance) tracks blooming-like motions of the extracellular domains, PC2 (30% variance) (Fig. 8a, left) describes quaternary twisting and pore gating (Fig. 8a, right), in accordance with the literature[51,55] (Supplementary Fig. 8a). Notably, structures are separated into functional clusters predominantly by PC2, which shows the strongest correlation with pore radius and contributes with up to 84% to the gating transition (Supplementary Table 6). The projections onto the PC1-2 subspace divide the ensemble into five well-defined clusters related to their functional status and crystallization conditions (Fig. 7c, left). Remarkably, the PCs split the locally closed as well as the open structures into two groups along PC1, which differ in their extracellular diameter (80 versus 70 Å) (Fig. 8b) suggesting two possible routes for gating: one 'bloomed' leading from 4NPQ to 4HFI, crossing a series locally closed structures (3TL*) stabilized by Cys-bridges[52] and a structure in equilibrium between locally closed/open (4NPP)[51]; the other route, with 'un-bloomed'

structures, leads from 4NPQ to the rightmost open cluster passing the locally closed 4LMJ and 4LMK. The fact that both groups distribute concentrically along a near diagonal axis suggests that they are functionally equivalent, and that the difference in compactness is due to their extracellular mobility in the crystal lattice. After examining crystallization conditions, we found that >90% of the bloomed structures were solved at a higher temperature (293K) than those un-bloomed at the right (277K). Only five outliers are found (Supplementary Fig. 8b): the open structures 4IRE and 4F8H, which appear in the high-temperature (bloomed) region but were solved at 277K, and the structures 4LMJ, 4LMK and 4LML, in the low-temperature (compact) region but solved at 298K. Interesting, the first two correspond to GLIC bound to ketamine (4F8H) and loop C mutations (4IRE) that actually inhibit the channel. Although they are open in the TM region (pH 4), these pentamers appear more bloomed than other non-inhibited structures at 277K. In contrast, the un-bloomed 4LM* locally closed and open structures are more compact than others at high-temperature. They harbour TM2-TM3 loop mutations designed to impair proton binding and gating: loss-of-function changes in the locally closed ones, and a double rescuer mutation in the open-like one[56]. This suggests that extracellular blooming at a given temperature may be a subtle signature of the channel functional status, and certainly deserves further investigation.

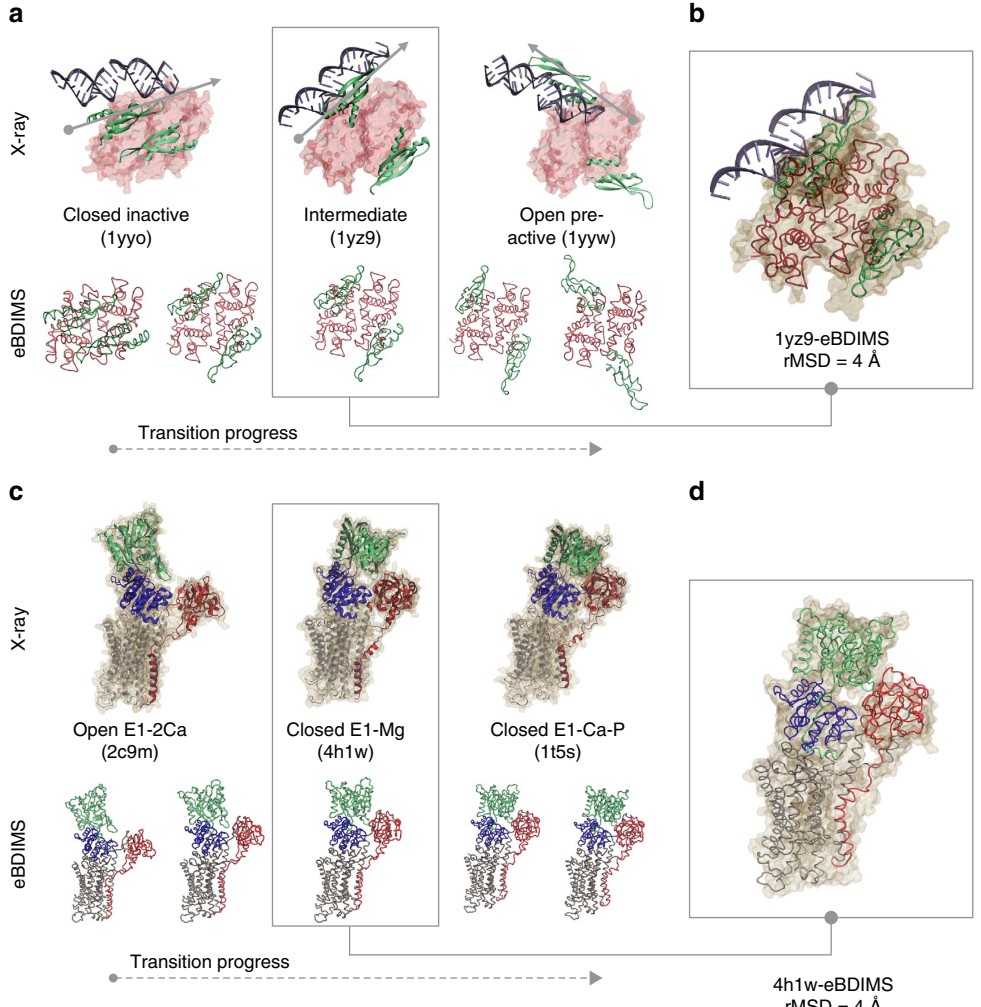

**Figure 6 | Comparison between the crystallographic and the eBDIMS pathways for RNaseIII and SERCA.** (**a**) Upper row: The complete X-ray structures for the RNaseIII end-states and the intermediate are shown bound to the dsRNA substrate; these conformations represent three sequential snapshots along the RNaseIII cycle. The starting structure is bound but catalytically inactive, and the following two show the progress to reorient the dsRNA onto the catalytic cleft (approximately in the z axis of symmetry). Lower row: as the two dsRNA-binding domains start to open the eBDIMS pathway shows how they spontaneously sample PC1 approaching the 1yz9 intermediate (shadowed); then, they leave it behind as rotation along PC2 progresses. (**b**) Superimposition between X-ray intermediate structure 1yz9 and the best overlapped eBDIMS frame. (**c**) Upper row: The complete X-ray structures for the SERCA open and closed end-states and one of the $Mg^{2+}$-bound visited intermediates (4h1w) are shown; the structure is approached within 4 Å (**d**) by the forward eBDIMS pathway tracking headpiece closing.

The projection of the eBDIMS opening trajectory onto the PC1-2 subspace shows a smooth sampling of the motions encoded in the X-ray ensemble, sequentially visiting the locally closed intermediates (Fig. 7d); the ordering of locally closed structures by PCs perfectly agrees with their rMSD from the end points. Un-blooming precedes the quaternary twist decrease as previously suggested[51,55] and in accordance with PCs (Supplementary Fig. 9a–e). Even in such a concerted multi-subunit transition, the motions are again encoded in low-frequency NMs (Supplementary Table 7). For GLIC, most methods are capable of sampling this small transition but nevertheless differ in their linearity when projected onto the PC1-2 subspace, with eBDIMS providing the broader sampling (Fig. 7c). There is once again only a subtle asymmetry of the reverse pathway (0.06; Supplementary Fig. 5): while the transition starting from resting GLIC proceeds through locally closed structures as the channel opens, the closing transition follows a path slightly left-shifted. Considering that bloomed locally closed and open structures are poorly separated by PC1, both pathways

appear essentially equivalent, suggesting reversible forward/reverse routes for gating in which blooming and quaternary twist motions proceed in concerted fashion. As for SERCA and RBP, MD simulations suggest that both pathways explore the same low-energy passage connecting closed and open GLIC (see below).

**Intermediates sampling in atomistic FELs.** To explore the significance of pathway divergences on the PC-subspace, the sampled intermediates and how they relate to the lowest energy paths between end-states, we computed FELs from atomistic MD for three cases: (i) RBP hinge bending; (ii) SERCA headpiece closing and (iii) GLIC cooperative gating. These examples have been previously studied with MD[57–59] and are known to reversibly transition between end-states in the absence of complex ligands, thus being suitable for comparison with eBDIMS. The atomistic methods to track these changes are also representative of common MD implementations (see Methods):

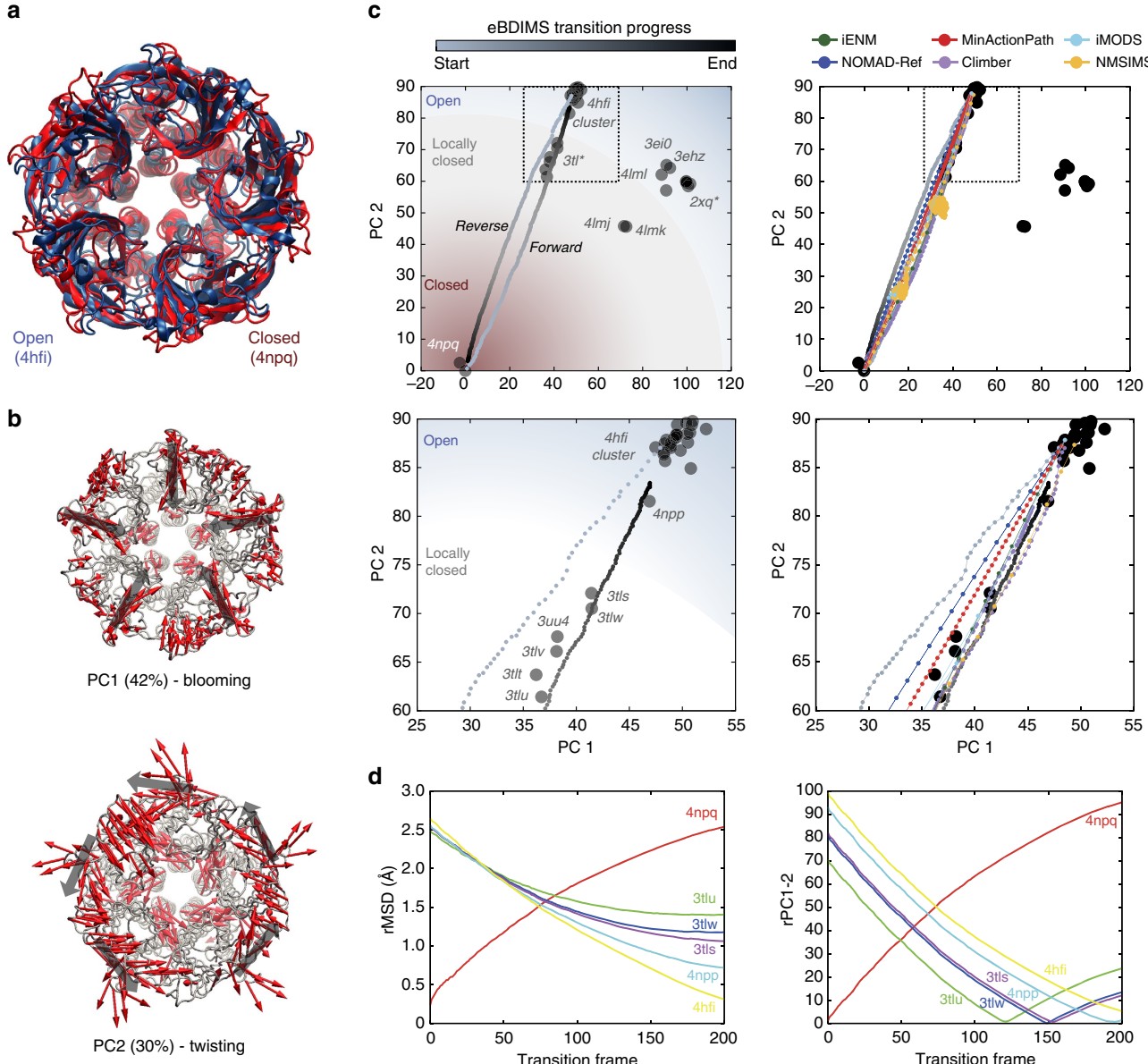

**Figure 7 | Conformational transition of GLIC. (a)** The two crystallographic structures representing the resting state (4npq) and the open conducting state (4hfi). **(b)** Dominant PCs of the GLIC X-ray ensemble (46 structures) versus 4npq. **(c)** Projection of the eBDIMS pathway onto the two major PCs of the ensemble (left); zoom highlighting the distribution of structures in the locally closed region (right). **(d)** Comparison between the forward pathways computed by eBDIMS, iENM, NOMAD-Ref, MinActionPath and Climber. Reverse pathways generated by eBDIMS, iMODS, NMSIM and Climber also shown.

biasing along a heuristic reaction coordinate (RBP); standard MD from unbound structures (5′NTase); multi-run MD simulations (SERCA) and a single microsecond-trajectory (GLIC). For RBP, we collected transition pathways from end-states with steered MD and AWH[28] using domain distance as reaction coordinate. In accordance with former studies[57], trajectories from the closed structure (2DRI) rapidly converge on open and partially open conformations, while simulations from the open (1BA2) overlap with the former but never reach the fully closed state in the absence of ribose. The FEL for the opening transition (Fig. 9a) reveals that the lowest energy through connecting end-state basins actually comprises most of the area delimited by eBDIMS forward/reverse paths; interestingly, experimental intermediates are found along the edges of this region, indicating that they correspond to meta-stable states captured by crystallization but not to energy minima in solution. For SERCA, several MD studies

have shown spontaneous $Ca^{2+}$-independent closing of the headpiece[58,60] in the absence of bound nucleotide. Specifically, a recent computational study[58] (also supported by FRET measurements) reported saltatory headpiece closure reaching long-lived states similar to the $Mg^{2+}$-bound structure (3W5B) here identified as an on-pathway intermediate. The E1-$Mg^{2+}$ configuration was suggested to approach an elusive E1-apo intermediate between E2-free and the E1-$2Ca^{2+}$ state, which would explain the accelerating effects of $Mg^{2+}$ on $Ca^{2+}$ binding[61,62]. According to the combined PCA/eBDIMS analysis, this configuration may as well represent a transient state in which $Ca^{2+}$ and ATP sites are pre-poised for efficient nucleotide binding. We compared the FEL sampled by MD[58] from the open cluster (1SU4) which show spontaneous closing, with the eBDIMS pathways from/to 1T5S. Once again, the forward/reverse routes sample the boundaries of a wide low-energy area for

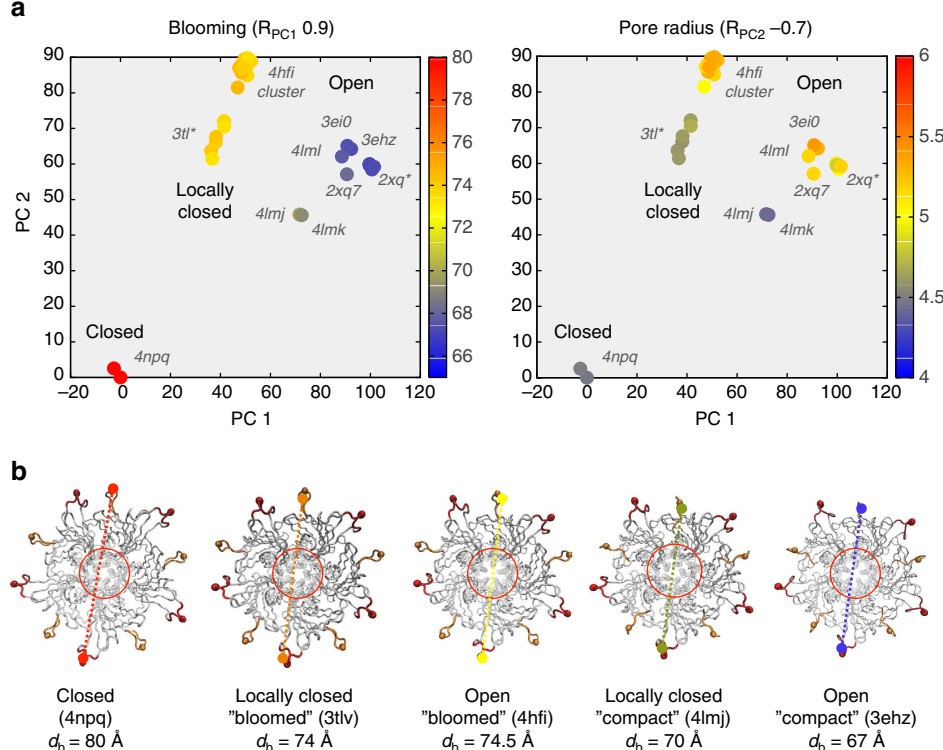

**Figure 8 | Correlations between structural variables for GLIC and the PC1-2 partitions reveal two possible temperature-dependent pathways for gating.** (**a**) Variation of major structural parameters in the PC1-2 subspace (Pearson Correlation in brackets; further details in Supplementary Fig. 8a). (**b**) PC1 partitions the locally closed and open structures into two subsets according to the expansion of the extracellular domain; these two subsets strongly correlate with the crystallization temperature (Supplementary Fig. 8b). Bloomed structures are solved at 290 K while the more compact on the right are mainly solved at 270 K.

the closing transition (Fig. 9b), populated by crystallographic intermediates. Finally, we compared the GLIC eBDIMS pathways with a microsecond-long simulation from the open state that spontaneously closes at pH 7 (ref. 59), and a shorter 500 ns from the closed state at pH 4 that evolves to the open state. Inspection of the corresponding FELs (Fig. 9c,d) suggests again that both eBDIMS pathways delimit the same low-energy route connecting the open/closed basins and thus are equally significant. Notably, as in the former cases, crystallographic locally closed intermediates do not fall into energy minima but rather sample the boundaries of the low-energy region sampled in the closing transition.

## Discussion

The aim of this work was to explore the possible reconstruction of protein transition pathways by prediction of intermediates from pairs of static structures. For that purpose, we developed a new sampling method, eBDIMS, and in parallel, a thorough PCA-based validation scheme to assess its biological relevance as well as to retrieve pathway information from structural ensembles. Although PCA has been used to extract motions sampled by MD[11], or to evaluate NMs from experimental ensembles[8], here it is applied to ensembles with enough sampled states to reconstruct the conformational landscape along meaningful CVs. We have shown that PCA of such ensembles automatically yields reaction coordinates that contain the one-dimensional system-defined parameters typically described for each protein. Moreover, the variance distribution informs on the dimensionality of the transitions: whether they are reducible to one (RBP or 5NTase) or a few coordinates (RNaseIII) or

rather need several (GLIC) to be fully described. The complexity of the PCs emphasizes the risk of biasing trajectories using simple variables, and also suggests that a deeper study of ensemble-PCs can help to understand how complex motions are coupled in transitions. Furthermore, structure clustering by PCs can distinguish not only functional states, but also specific experimental conditions. For the GLIC ensemble, PCA clustering detected a previously unnoticed partition of the solved structures into two groups dependent on the two crystallization temperatures used for solving them (277 and 293K). The fact that the only outliers in both groups are either bound to molecules or harbour mutations to perturb channel activity suggests a possible link between intrinsic mobility of the extracellular piece and the channel status which is reflected in its compactness in the crystal lattice. Thus, the PCA method by itself can help to assign a functional status to new structures, understand how they relate to each other or even raise experimentally testable hypotheses.

More important, PC-clustering automatically provides the most probable sequence of structures along a transition avoiding uncertainties introduced when dealing with several heuristic variables in complex transitions such as GLIC. By extracting the common pattern of motion not for a pair but for all representative conformations for a protein, the PCs uncover the ensemble-encoded routes for conformational changes. Using the PC1-2 bidimensional space as reference for benchmarking, it is also straightforward to evaluate how well sampling algorithms explore the conformational space. The projections onto the major PCs clearly distinguish feasible trajectories, characterized by a smooth and stable sampling of the experimental motions that spontaneously approaches intermediate states. On the basis of

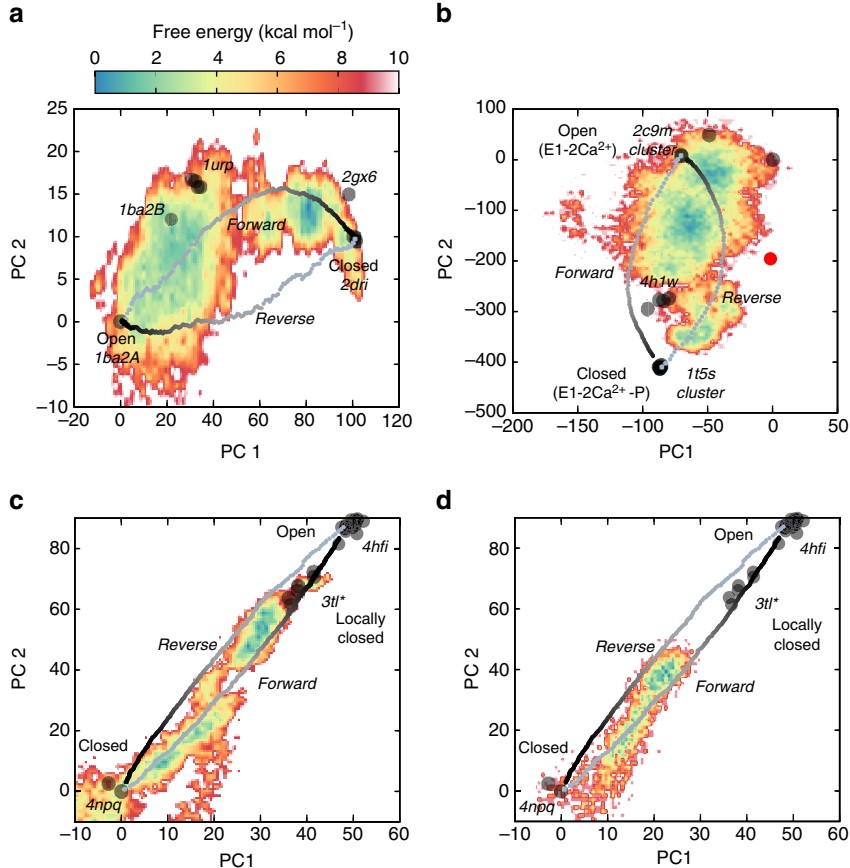

**Figure 9 | Overlap of eBDIMS forward and reverse pathways with free energy landscapes obtained from atomistic simulations.** (**a**) RBP opening with biased MD trajectories (1 μs); (**b**) SERCA headpiece spontaneous closing in multi-run MD simulations from the open conformation 1su4 (2 μs) reaching the intermediate state cluster; (**c**) GLIC spontaneous closing in a single long 1 μs-simulation at pH = 7 and (**d**) opening in a 500 ns simulation at pH = 4.6. Note that the slightly divergent pathways enclose most of the lowest energy passage between the end-states, with the approached crystallographic intermediates distributed at the boundaries.

this stringent evaluation, we demonstrate that it is possible to reconstruct the structural landscape and predict possible intermediate structures using path-sampling algorithms. Notably, the CG eBDIMS trajectories greatly overlapped with the atomistic method Climber, based on pulling and energy minimization of the force-field ENCAD. The ability of a pure ENM topology-based simulation such as eBDIMS to predict intermediate states is a strong demonstration of the emerging paradigm that the large-scale functional dynamics of proteins is greatly encoded in their overall shape[32] and does not depend on fine-grained sequence details.

While in some proteins (for example, RNaseIII) apparently there is one topologically accessible sequence of motions that allows for a transition, in others (for example, GLIC) several concerted motions proceed simultaneously until completion. Our results suggest that the ability to spontaneously sample a transition is pre-encoded in the structures, not only for ligand-free forms[63] but throughout the conformational change. We observed that the NMs were well aligned with either the distribution of structures around each basin, and/or with the actual direction of changes defined by the nearest intermediates. It has been common practice to evaluate the significance of NMs by their overlap with the difference vector between conformations. Although it is a fair approximation for most proteins, it can fail when the experimental path follows an intermediate that departs enough from both end points (Supplementary Fig. 4). Similarly, it becomes clear why following the modes better overlapped with the transition vector can render

distorted pathways as such motions are often not sampled experimentally. Nevertheless, most ENM-based algorithms perform extremely well providing fast approximations to the experimental paths in simple cases (see Supplementary Note 1).

Interestingly, both Climber and eBDIMS show nearly identical divergence in the forward/reverse pathways; however, considering PCs variance, this asymmetry is small and only for some examples hints to distinct sequences of motions as a protein progresses along its functional cycle. The latter is notable in the hysteresis-like cycle between RNaseIII RNA-bound structures (Fig. 4c), with nearby intermediates in both directions. In this extremely complex transition, eBDIMS and Climber suggests asynchronous motions (RBDs opening/rotation in the forward route, followed by closing/rotation in the reverse). Here pathway divergence, already evident from the PC1-2 distribution of experimental conformers, contains information about the sequence of movements in a multi-step complex landscape; unbiased MD from unbound forms (Supplementary Fig. 6c) also hints to differences in the preferred forward/reverse paths as suggested for some proteins[1]. However, when asymmetry is low, comparison with the FELs from MD rather suggests the equivalence of forward/reverse routes: in the three cases examined, they appear to delimit not different linear paths, but rather an area in the conformational landscape that overlaps with the lowest energy passage connecting the end-state basins. Notably, the crystallographically trapped intermediates in all cases tend to distribute along the edges of these

regions, pointing that they correspond to meta-stable states but not to energy minima in solution. Apparently, transitions from open (enthalpy-driven) and closed (entropy-driven) forms provide two alternative but energetically feasible solutions for a path, indicating the usefulness of non-linear methods to explore lowest energy channels in the protein landscape.

With the rise of cryo-electron microscopy (cryo-EM)[64] and time-resolved X-ray techniques[65] capable of trapping structures in several conformations at once, the proteins solved in three states or more will no longer be anecdotic but rather standard. Overall, the methodology here outlined offers a powerful approach to rationally categorize as well as extract dynamic information from experimental ensembles, expand them by computation of feasible intermediate states and suggest the lowest-energy passages connecting them, thus paving the way for an intelligent exploration of the conformational space.

## Methods

**Model ensembles and intermediate definition.** We selected proteins that undergo large conformational changes (Supplementary Table 1), and for which ensembles containing at least three clearly distinct conformations are available, allowing the computation of robust PCs to neatly cluster onto functional states; all these examples but GLIC were previously studied by Weiss and Levitt[4]. We consider those ensembles as 'structurally rich' in terms of sampling, regardless of the number of structures present. This selection criterion is different from all previous studies, focused on the quantity of structures available for a protein rather than the actual sampling covered by them. In practice, it means that each ensemble: (i) has at least three different well-defined functional clusters in the PC1-2 subspace and (ii) the first PCs have a significant weight describing distinctive large-scale motions that cover >70% of the structural variance. We demonstrate here that such ensembles provide extremely robust PCA, being possible to reproduce >70% of the complete ensemble motions and variance distribution with just three structures sampling the transition path that is, one for each end-state and one intermediate (Supplementary Table 2 and Supplementary Fig. 1); robustness is higher as larger is the scale of the conformational change and the variance accumulated in the first PCs.

In order to retain the maximal structural information and perform full-length PCA, we selected near complete structures sharing 95% sequence identity. Structures with less than five missing residues and not targeting hinge regions were repaired and this residue limit was extended only when reconstruction did not perturb the PCA results (that is, modelled structures fell into already populated clusters); this condition excluded myosin from the original Weiss benchmark. The systems selected range from a middle sized protein such as RBP (271 residues) to the large GLIC pentameric assembly (1540 residues), and their conformational changes involve very different types of rearrangements, from subtle cooperative motions in GLIC (rMSD = 2.6 Å) to large-scale complex rigid-body rotations and translations for RNaseIII (rMSD = 18 Å) or hinge domain bending and twisting in RBP (rMSD = 6 Å).

For each protein, the starting structure (state 1), used as reference for ensemble alignment, was defined as that in the inactive, resting or not-stimulated state (that is, not bound to ligand, substrate, signal and so on) as opposed to a target or active conformation (state 2; Supplementary Table 1) in order to facilitate the interpretation of projections as deformations with respect to a real structure instead of a geometric average. Once the ensembles and the end-structures were defined, we computed transition pathways with eBDIMS and seven other path-sampling methods and performed PCA to evaluate them (see below).

Intermediate states were defined primarily a priori on the basis of the literature for each system or a posteriori after examination of ensemble and pathway projections. Thus, we distinguish two types of pathway intermediates (Supplementary Table 1): (i) knowledge-based (that is, heuristic) intermediates, which are characterized as such and have been trapped by specifically designed mutations and so on; (ii) topology-based intermediates, which appear as natural conformers along the routes connecting end-states in the PC1-2 subspace, and correspond to structures bound to inhibitors or harbouring mutations that have accidentally trapped a transient conformation.

**Essential dynamics-based ENM (ED-ENM) driven Brownian DIMS (eBDIMS).** The eBDIMS approach implements the MD-derived nearest neighbours ED-ENM potential[10] in a Brownian dynamics simulation[66] to trace physically feasible trajectories from a starting state, $R_0$, to a target state, $R_t$ (Supplementary Fig. 10). The protein is considered as a network of $C^\alpha$ particles connected by springs, moving randomly in a stochastic bath. The equation of motion for each particle follows the Langevin equation:

$$m_i\ddot{r}_1 = F_i - \gamma\dot{r}_1 + \xi_i(t), \tag{1}$$

Where each residue i is represented by the coordinates of its α-carbon ($r_i$) and has a mass of 100 Da (average aminoacid mass). The second term is a dispersive force, accounting for the viscous resistance that the particle feels on going through the fluid (given by friction coefficient $\gamma$) whereas $\xi_i(t)$ is a white noise vector that accounts for fluctuations due to the thermal motion of the solvent. The random force given by the stochastic process $\xi_i(t)$ satisfies two conditions: first, is Gaussian with zero mean, and second, its autocorrelation function has the form (see further details in refs 29,66):

$$\langle\xi_l(t).\xi_n(t')\rangle = 2mk_BT\gamma\delta_{ln}\delta(t-t') \tag{2}$$

Where $k_B$ is the Boltzmann constant, $T$ is the temperature of the stochastic bath (300K), and the δ-Dirac functions ensure the independence of the components of the noise vector. Besides of representing the solvent, the friction and noise terms create a natural thermostat where random energy shots are balanced by the dissipative forces, keeping constant temperature and energy.

The stochastic equation of motion in (1) is integrated numerically with the Verlet algorithm, which gives for the velocities and positions after timestep $\Delta t$ (1 fs). On the above equation, the force acting on each residue i, $F_i$, is computed assuming hookean elastic potentials for its interactions with the rest of residues j:

$$U_i = \frac{1}{2}\sum_{j=1}^{N} K_{ij}\left(r_{ij} - r_{ij}^0\right)^2, \tag{3}$$

Where $N$ is the number of protein residues, $r_{ij}$ and $r_{ij}^0$ are the instantaneous and equilibrium distances between pair residues i and j, and $K_{ij}$ their spring constant defined by the ED-ENM force-field which sets fixed MD-calibrated values for the first three C-alpha neighbours in the peptide chain to keep backbone stereochemistry, and a exponential function for long-range non-sequential interactions (details in ref. 10).

All the parameters in the ENM potential[10] and the BD simulation engine[66] were carefully optimized to reproduce the sampling by standard force-fields using as reference the MoDEL (molecular dynamics extended library)[67] database of state-of-the art atomistic simulations with explicit solvent, as well as experimental data from X-ray crystallography and NMR. At the default temperature of 300K, the friction force is balanced to act as a thermostat according to the fluctuation-dissipation relation (see above). Note that the random forces acting on each particle (set by the random seed) render minimally different paths each time (Supplementary Fig. 11a).

Biasing of the trajectory in the direction of the transition is achieved by dynamic importance sampling (DIMS) based on an informational criterion[68,69] where a Maxwell demon is introduced to enrich the trajectory in movements that approach the structure towards the target. Accordingly, for every certain number ($k$) of unbiased cycles, a progress variable, $\Gamma_i$, for the instantaneous structure, $R_i$, is recomputed and compared with the target one, and used as criteria to accept or reject the random moves (see also ref. 42). Here we define the progress variable in terms of the difference in pairwise distances of the starting and target structures:

$$\Gamma_i = \sum_{i,j=1}^{N}\left(d_{ij} - d_{ij}^0\right), \tag{4}$$

Which is compared every $k$ steps to that of previous step, $\Gamma_{i-1}$, so that the current conformation is accepted if decreases its value, or rejected otherwise.

The iteration proceeds until convergence into the target basin is achieved, that is, the sampled structures reach an rMSD with the target in the range of thermal oscillations (within 1–3 Å depending on system size and amplitude of the conformational change). Trajectories tend to converge even when started from different points along a pathway, for example, from different structures belonging to the same cluster (Supplementary Fig. 11b). Although all parameters are optimized to work with default values for any protein system, a wider or fastest sampling of the conformational space can be achieved by modulation of: (1) the number of unbiased steps $k$, (2) the cutoff for ED-ENM force-field long-range interactions, respectively; Specifically, increasing $k$ provides slower but wider sampling (useful for example, to fit experimental data) while decreasing it accelerates calculations but renders paths closer to a Cartesian interpolation (Supplementary Fig. 11c). The eBDIMS code can also run without a bias to populate the conformational space as described in refs 66,70.

Thanks to the ED-ENM potential, eBDIMS method can work with limited information from the target structure that is, a very small set of distance restraints. Since the ED-ENM algorithm sequentially assigns the force constants to the three nearest neighbours along the peptide chain, the algorithm works even when the target structure has large missing regions, given that a complete starting reference is provided (see example for SERCA in Supplementary Fig. 7a).

**Transition analysis by PCs projection.** PCA[6] is a statistical technique to reveal dominant patterns in noisy data. The diagonalization of the covariance matrix of the system allows for obtaining the major axes of statistical variance or PCs. In this way, complex multidimensional data is mapped to a reduced set of coordinates, which contain the dominant trends explaining their variation. PCA has been widely applied in structural biology to analyse structural ensembles. Protein structures are aligned to a reference in order to compute the covariance matrix, which describes the mean-square deviations in atomic coordinates from their mean position (diagonal elements) and the correlations between their pairwise fluctuations (off-diagonal elements). Diagonalization then yields a set of eigenvectors and eigenvalues representing the motions that explain the variation in the atomic coordinates. In the structurally rich ensembles analysed here, the first two components contain at least 70% of structural variation of the ensemble and

are very robust even when considering a minimal number of structures (three) as discussed above (see Supplementary Table 2). Within this framework, any structure $i$ is characterized by its projections onto the conformational space defined by the two major components, $PC_k$:

$$PC_k = |T_{i-0}| \cdot \cos(PC_k \wedge T_{i-0}), \qquad (5)$$

Where $T_{i-0}$ is the difference between the coordinates of $i$ structure and the chosen reference, and $PC_k$ is one of the major axis. Since we selected ensembles with structures sampling not only end states but also known intermediates, these major PCs actually describe the pathway for conformational change, and projection onto the new coordinates reveals the ordering of structures as a transition proceeds. The comparison between distances in rMSD and in the PC1-2 subspace to intermediates and target structures for the methods tested reveals subtle but relevant differences between the two measures (Supplementary Table 5). The Euclidean distance in terms of the PC coordinates provides an alternative metric, which weights the differences between any two given structures according to the relevant motions thus filtering out local fluctuations that contribute to the rMSD. Note the clear detection of intermediates in Figs 2–5 and 7, panel d, which is more pronounced than in the equivalent rMSD profiles.

**Comparison to standard path-sampling algorithms.** The eBDIMS trajectories are compared with other ENM-based methods of different complexity: NOMAD-Ref[44], which uses ENMs to interpolate interresidue distances with the algorithm of Kim et al.[39]; MinActPath[45], which solves analytically the Langevin equation for harmonic potentials at each side of the transition and finds numerically the crossing points of the solutions; finally, the iENM[46] is based on solving the saddle points of a double-well potential by linearly interpolating between the end-states ENM potential functions while iMODS[20] interpolates in the dihedral angle space. We also compare with the non-linear atomistic algorithms NMSIM[47], which uses a complex three-steps procedure, and Climber, based on the molecular mechanics ENCAD potential[4]. All the methods were run with their default parameters (See Supplementary Table 4 and Supplementary Methods).

For the non-linear eBDIMS and Climber, trajectories were computed in both forward and backwards directions; and their asymmetry score (ranging from 0 to 1), evaluated as the eccentricity of the resulting ellipsoids in the PC1-2 subspace; the rMSD contour plots between the forward/reverse trajectories were also computed (see Supplementary Methods and Supplementary Fig. 5). Note that, in the PC1-2 subspace, a Cartesian interpolation such as that provided by the MolMov server (Morph)[19] projects as a straight line between the end-states regardless of the energy minimization of the structures (see summary of methods in Supplementary Table 4). To gain further insight into how motions are imprinted in the structures, we also created ensembles along the lowest frequency NMs[10] with a simple Monte Carlo routine.

**MD simulations.** Crystal structures of RBP (1BA2, 2DRI, 1URP), RNAseIII (1YYW, 1YYO, 1YZ9), 5-NTase (1OID, 1OI8, 1HPU) and GLIC (4NPQ) were used for MD simulations. Each system was solvated with TIP3P waters, energy minimized and equilibrated, and production runs were carried out with no restraints under isothermal-isobaric (NPT) ensemble. Steered MD and AWH simulations of RBP were run under the same conditions as the production runs, with the same reaction coordinate (interdomain distance between center-of-masses). See detailed protocols in Supplementary Methods. Dr Seth Robia and Dr Marc Baaden generously provided trajectories for SERCA (1SU4) and GLIC (4HFI), respectively. The SERCA simulations were performed in explicit water under NPT conditions ($T = 300K$) as described in detail in Smolin and Robia[58]. GLIC simulations were also performed in explicit water under NPT conditions as described in Nury et al.[59]. A list of all simulations and their overlap with eBDIMS is provided in Supplementary Table 8.

**Calculation of heuristic system-defined structural variables.** The heuristic system-defined variables for each protein were computed according to the specific literature using in-house tools combined with Visual Molecular Dynamics (VMD) scripts (see details in Supplementary Methods).

**Data availability.** The original FORTRAN code for eBDIMS and source data for all figures and tables is available upon request to the authors.

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

## Acknowledgements

This work was funded by IRB (M.O., O.C., L.O.), the Swedish Research Council (2013-5901) and the Swedish e-Science Research Center (E.L, L.O., O.Y.). The authors acknowledge the use of computational resources by the Swedish National Infrastructure for Computing (2015/16-45). L.O. was supported by a postdoctoral scholarship from SeRC/Vetenskapsrådet. L.O. thanks Dr Johan Gustavsson for helpful discussions and suggestions, thorough reading of the manuscript and help with the preparation of figures and interactive plots. Thanks to Spanish Ministry of Science (MINECO) grant Bio2015 64802-R, Catalan AGAUR, European H2020 Program (Bio Excel CoE) and European Research Council (ERC Advanced Grant SimDNA). M.O. is an ICREA Academia Fellow. We thank Dr Seth Robia and Dr Marc Baaden for generously providing SERCA and GLIC MD trajectories.

## Author contributions

L.O. conceived the original idea, developed the eBDIMS code, performed, analysed and interpreted calculations and wrote the paper. O.Y. performed PCA and eBDIMS calculations, and MD simulations. O.C. wrote the initial eBDIMS code. M.O and E.L. critically read the manuscript and helped with useful suggestions for calculations and data interpretation.

## Additional information

**Competing financial interests:** The authors declare no competing financial interests.

