## [Peer review file · Nature Communications]

Reviewers' Comments:

Reviewer #1 (Remarks to the Author)

In this manuscript, the authors present a path sampling algorithm based on a novel combination of elastic Network model (ENM) with Brownian simulations. The method, called eBDIMS, is validated with four very well documents cases. The paper is well written and it reads easily, I only have some issues that I'm reporting hereafter.

Major:

- The method is tested on only four cases. The authors claim that they are "highly" diverse systems. They are four different cases but I am wondering how representative are they? Ideally, they should enrich the validation test with more test cases (at least all the examples used in [4]). They could also explore the transitions observed in very long simulations (see J. Chem. Phys. 139, 121912 (2013))
- They compare they results with several linear methods such as NOMAD-Ref (2006) or MinActPath (2007). However, there are more recent approximations based on similar principles many of them available as web servers. Among others ANMPathway [2], iMODS[21], and NSIM [46] can be easily added to your comparison.
- Could you comment in more detail the pros and cons of eBDIMS respect to Climber?
- Just a curiosity, what happen when you start from an intermediate (e.g 1yz9 instead 1yyo)? It will follow the same open-close path?

Minor:

- The definition/characterization of a "real" intermediate is at least controversial. The authors should add a comment on this.
- One of the strongest points of eBDIMS should be the efficiency. They report that is fast, but how long it takes? Is it faster than climber?
- Why iENM and NOMAD-ref cannot track the transition in fig 4 C? Why the trajectories are so jagged?
- Figure 1. The alignment of open-close trajectories is really hard to see.
- Figure 2. In panel A, it is the same structure.
- Some of the arrow representations are too dense and it is difficult to grasp the overall motion. Could you reduce the number of arrows to improve the figures?

Reviewer #2 (Remarks to the Author)

A. The overall aim of this work is to predict and validate protein structures that lie between the known endpoints of conformational transitions. To this end, the authors have performed Langevin dynamics and compared the trajectories from their simulations against known structures in between the end points. The figure showing this comparison has been placed in the supplemental material, and yet this is one of the most important outcomes. Interestingly they have performed simulations in the two different directions between the endpoints and importantly observe different pathways in the two directions. The transitions from closed to open forms have always been more difficult to achieve with such elastic models so this results showing transitions in that direction are important.

B. The novelty of the work lies primarily in the Langevin simulations, and in using them to perform simulations in two different directions. But they have not been sufficiently described, and it remains unclear whether the parameters involved have been sufficiently tested and validated.

There are a number of important points, however, that the authors have not addressed in the present manuscript:

C. and D. How reliable are the PC's? There is the important issue of whether the set of available structures is sufficient for developing reliable PC's. Eleven and sixteen structures for two of the structure sets used here are very small numbers. In our past experience such small sets can yield significantly distorted views of the conformational space. A recent paper in J Chem Phys investigated the convergence of PCs using different numbers of structures, which should be done here. But, it also might be possible to devise weighting schemes for the structures.

The Langevin simulation parameters have not been sufficiently described. Replicas of the simulations have not been performed to validate the trajectories.

E. The energetics have been completely ignored, with only geometry and entropy being considered. If energies were considered the trajectories might be significantly distorted. The authors could also have utilized atomic elastic models which would have a different set of potential functions from the coarse-grained cases. This omission of energies raises serious questions about the reliability of the results.

In general there must be some energy-entropy compensation along the trajectories since the closed forms would be expected to have lower energies and the open forms higher entropies. In fact these differences may account for why the trajectories differ in the two directions.

F. Summary of the problems with the paper. The PC's require some validation to justify the use of such small numbers of structures. The work needs to describe in more detail the simulations and how the parameters for them were chosen. Replicas of the simulations need to be performed to observe their variabilities. Adding some evaluation of the energies is also critical to discern whether these trajectories lie in two separate free energy troughs or how these particular trajectories relate to the details of the free energies.

G. There are a number of additional references that could have been cited.

H. The paper is clearly written.

Reviewer #3 (Remarks to the Author)

The work by Orellana et al. contains two attractive approaches: first, for classifying multiple structures of proteins (or protein complexes) within a single conformational landscape; second, for determining probable transition pathways between distinct structural groups. As more and more proteins are solved in different conformational states, these approaches will find many applications and should help to understand the functional mechanisms of the proteins in question. The methods that have been developed are clearly described and their application to four well-studied cases is nicely documented and gives rise to some interesting results.

My main queries involve the automated application of these techniques to structural data, particularly when little or nothing is known about the function of the proteins studied:

1. Why is it necessary to use an identified apo structure as the reference in the PCA analysis? I can understand this choice in the case of elastic network models where the contact network of "closed" structures makes closed-to-open transitions more difficult than the reverse, but what would change if the PCA analysis used another structure as the reference? In the future, it is probable that cases will arise where the apo/holo distinction is not known a priori.

2. How robust are the PC's to the removal of one or more structures within a given structural

group?

3. I did not understand the definition of the progress variable γ . What is meant by the "internal distances" d_{ij} and how do they direct the dynamics to the target structure?

4. Do any modifications have to be made to the eDIMS procedure for treating different protein structures? For example, the number of unbiased steps " k ".

Apart from these points, I think this contribution will interest both structural biologists and modelers concerned with the functional behavior of proteins.

Reviewer #1 (Remarks to the Author):

In this manuscript, the authors present a path-sampling algorithm based on a novel combination of elastic Network model (ENM) with Brownian simulations. The method, called eBDIMS, is validated with four very well documents cases. The paper is well written and it reads easily, I only have some issues that I'm reporting hereafter.

We are very pleased with the reviewer's positive response, and we have done our best to address the issues raised. Below, we provide our answers point by point.

Major:

*1.1 - The method is tested on only four cases. The authors claim that they are "highly" diverse systems. They are four different cases but I am wondering how representative are they? Ideally, they should **enrich the validation test with more test cases (at least all the examples used in [4]).***

Although the selected proteins are diverse in function and shape, we certainly agree that only four cases make a limited dataset. Nevertheless, the structural data available for these four proteins is exceptional. The benchmark by Weiss and Levitt¹ included five proteins that all have structural intermediates along a large conformational change (5NT, RBP, RNaseIII, myosin, and SERCA). We explored three of them in our initial manuscript (5NT, RBP and RNaseIII), including GLIC because of our past experience with the system and its rich structural information. Upon submission, we excluded SERCA and myosin for two different reasons:

- 1) The **number of myosin structures is limited** (only 12 near full-length), but more importantly, they **contain several large gaps of missing residues, some located at hinge regions for the transition**, which prevents its use for a reliable PCA as well as for pathway sampling. We have now mentioned this in the manuscript.

*See: **Methods - Model proteins, ensemble quality and intermediate definition***

- 2) Upon elaboration of the original manuscript, we noticed that **the SERCA transition (1SU4 ↔ 1IWO) as well as the SERCA intermediate (1VFP) suggested by Weiss & Levitt did not seem feasible considering the progression along the catalytic cycle and specifically, its mapping onto the PC1-2 subspace (Fig.5)**. Instead, our combined PCA-eBDIMS study of SERCA identified the cluster of **E1-free states around 4H1W (Fig.6) as well as the 4NAB structure**, found when using eBDIMS for reconstruction (see *Fig.S6A*), **as possible intermediates**. These "novel" intermediates are **supported by recent FRET data and MD simulations² (Fig.9)**. We are currently preparing a separate manuscript on further SERCA transitions, but to be consistent with the benchmark in¹ we have decided to include again part of our findings in the revised version of the manuscript.

See: Results - Sarco-Endoplasmic Reticulum Ca²⁺-ATPase (SERCA), and related figures (Fig.5, Fig.9, Fig.S1-4 and S6), and Discussion

If we oversee the condition that the ensemble proteins have transition states characterized as such in the literature and extend the definition of intermediate (*See point 1.6 in minor issues, where we discuss this*) to any structure visited along a PC1-2/eBDIMS pathway (trapped by mutations, ligands, etc), many further examples can be found in the Protein Data Bank (PDB). **We are developing a curated database** of such cases; we include five of them currently under study at the end of this letter (*Fig. A*).

1.2 - They could also explore the transitions observed in very long simulations (see J. Chem. Phys. 139, 121912 (2013))

We also appreciate the reviewer's suggestion to include long-timescale simulations. We have carefully studied the two examples with ms-long Anton trajectories in the paper by the Bahar Lab³. Unfortunately, **the carnitine transporter has only two solved structures, which prevents the calculation of PCs** to compare Anton simulations and eBDIMS paths in an independent set of coordinates (*at least three structures are needed, see points 2.4 and 3.2*). On the contrary, **BPTI has a large number of structures, but given the small size of the protein and reduced scale of motions, the resulting PCs are not robust** enough to neatly cluster the conformations onto functional states, being extremely sensitive to minor changes in ensemble composition, for example, when including/excluding NMR structures as well as the clusters explored in the ms-Anton simulation⁴ (see *Fig.B* attached at the end of this letter). As we discuss in the paper, being a coarse-grained method, **eBDIMS is suited to address collective conformational changes and preferably large systems**, rather than small proteins where local side-chain fluctuations define structural variance; in our opinion, MD simulations address these cases much better.

However, we totally agree with the referee that it is extremely interesting to show how eBDIMS and long-timescale MD compare when sampling conformational transitions, and we are actually doing it in our lab with several studies of ion channels and pumps. To show the utility of combining the PC1-2 subspace with eBDIMS to monitor MD simulations, **we have now included comparisons with microsecond-long trajectories for RBP, SERCA and GLIC**. As can be seen, the eBDIMS paths greatly overlap with the MD-explored transitions (*Fig. 9*); in fact, **we have found that the forward/reverse pathways seem to delimit the edges of low-energy passages that connect end-conformations** (*See point 2.8 raised by Referee #2, where we further elaborate on this*).

See: Results - Sampling of intermediate states in atomistic free energy landscapes, and related Figures (Fig.9), and Discussion

1.3 - They compare their results with several linear methods such as NOMAD-Ref (2006) or MinActPath (2007). However, there are more recent approximations based

on similar principles many of them available as web servers. Among others ANMPPathway [2], iMODS [21], and NSIM [46] can be easily added to your comparison.

We also agree with the referee that it is interesting to compare several methods, especially if they are based on different physical assumptions (non-ENM or Langevin based), although our goal was not to rate them but to show how they explore transition pathways and how benchmarking against experimental data can help in algorithm development and intermediate identification.

As suggested, we have now included **NMSIM** and **iMODS** in our comparison, finding that they share the advantages (speed) and drawbacks (instability) of other simplified methods (*Figs. 2-8*) and provide very similar results. It must be noticed that the **ANMPPathway** server only returned results for the smallest transition, GLIC. We searched for additional servers but they cannot always deal with oligomeric structures (this also includes our previous webservers, Go-dMD⁵ and MD-dMD⁶) or are apparently no longer maintained (for example, FRODA).

See: Results and related figures (Figs. 2-8), and Discussion

1.4 - Could you comment in more detail the pros and cons of eBDIMS respect to Climber?

As mentioned above, all the tested methods are computationally very efficient and extremely fast compared with MD. The **clearest drawback of eBDIMS versus Climber is that, in its current implementation, it excludes the atomistic sequence-dependent details**; while its greatest advantage is that **it is an actual simulation, which allows for a greater versatility and sampling width, for example:**

- To provide **simulation-like rather than linear trajectories between end-points when decreasing the biasing rate**: this option, although slower, samples a much wider conformational space than any other of the examined approaches (useful for model fitting to experimental data from SAXS, NMR restraints, etc) (*Figs. S10C, k=1000*); in our lab, we are currently using eBDIMS sampling to fit SAXS data for SERCA (Dr. Magnus Andersson)
- To **work with a minimal distance restraint set from the target** thanks to ED-ENM coarse-graining, which allows reconstructing a conformer with missing gaps from a full-length template (practical example for SERCA in *Fig. S6A*)

Besides these, the eBDIMS algorithm, which was **developed from unbiased Brownian Dynamics, can also perform all its simulation functions** (as we reported in⁷⁻⁹), for example:

- Run unbiased trajectories along the initial linear pathways to provide further sampling,*

- ii) *Approximate the directions of conformational change when the target is unknown, due to the intrinsic sampling along the normal modes,*
- iii) *Include biasing from short MD-simulations,*
- iv) *Include other coarse-grained ligands/proteins (Dr. Carrillo, unpublished)*

We are currently **developing an improved version of the eBDIMS code for its implementation in a web-server** that will include more simulation-like features (*See also point 1.7, minor issues*); use of finer-grained resolutions is also possible but as discussed below (*See point 2.7 raised by Referee #2*) does not provide any improvement. Regarding the efficiency of each method, please see the discussion below in the section concerning minor issues (*See point 1.7*).

See: Methods and Figures S6 and S10

1.5 - Just a curiosity, what happen when you start from an intermediate (e.g 1yz9 instead 1yyo)? It will follow the same open-close path?

We consider this an excellent suggestion by the referee. Since the ENM force field is topology-based, we expected that the trajectories should follow a very similar path. We checked this for the easiest (RBP) and the most challenging (RNaseIII) examples, confirming that indeed, **trajectories launched from/along different points of a path or even between different end-states of the same cluster keep converging** and thus are very robust (see *Fig. S10B*).

See: Figure S10

Minor:

1.6 - The definition/characterization of a "real" intermediate is at least controversial. The authors should add a comment on this.

The referee raises an important point. In the initial manuscript we assigned the functional status (i.e. end-points and/or intermediates) by relying on the original research papers concerning each protein. We call this a *knowledge-based* definition. For example, for both GLIC and RBP, intermediate conformers were trapped introducing Cys-bridges to lock the protein along its functional cycle. Interestingly, we noticed that many structures not clearly identified as intermediates in the literature appear distributed along the transition paths explored in the PC subspace (for example, those identified for SERCA mentioned above, *point 1.1*), indicating a non-described but topologically-feasible intermediate. It has been suggested that different ligands or crystallization conditions can effectively trap intermediate states for proteins. To account for this situation, we have also introduced a PC/eBDIMS based definition of intermediate states in parallel to the knowledge-based one in the *Methods* section (see some examples of the first in *Fig.A*).

See: Methods and Fig.A at the end of this letter

1.7 - One of the strongest points of eBDIMS should be the efficiency. They report that is fast, but how long it takes? Is it faster than climber?

Since we cannot fairly evaluate the speed of web-server calculations, we have not included any time comparisons in the paper, but it should be noticed that **all the methods tested share an extreme computational efficiency, providing transition paths in very short times** (from a few minutes to hours in the most challenging case, RNAselll); the major differences are found in how they sample the experimental conformational space. In terms of speed, we were able to compare Climber and eBDIMS executables and found that they perform similarly, with a **slight advantage of eBDIMS for large systems like GLIC (5 versus 6h), and of Climber for large-scale motions like that of RNAselll (1 versus 2h), but neither of them rate among the fastest codes such as iMODS**, that provides transition paths in few minutes. The difference among them is mainly due to Climber running faster thanks to pulling when the random fluctuations explored by eBDIMS become a computational bottleneck, while eBDIMS outperforms Climber for large systems due to coarse-graining. Both codes can increase speed by increasing biasing frequency, but at the cost of approaching Cartesian-like straight paths (see Fig. S10C). In the present manuscript, our goal was to explore accuracy in intermediate prediction and thus we did not work on eBDIMS code speed optimization. We are currently **developing an improved version for its implementation in a web-server**, that takes full advantage of structure coarse-graining to accelerate calculations and will incorporate additional simulation-like features, as mentioned above (point 1.4).

1.8 - Why iENM and NOMAD-ref cannot track the transition in fig 4 C? Why the trajectories are so jagged?

We briefly commented on this question in the previous version of the paper. Our results suggest that **ENM-based methods may fail when choosing the biasing normal mode, by i) following the one that better overlaps with the transition vector instead of the one pointing to the next intermediate state** (which can depart from a straight interpolation between end-structures) or ii) **just following one of the lowest frequency ones** (which at certain regions can point to orthogonal directions in the conformational space, as seen for RNAselll and SERCA) (Fig.S3).

See: Fig.S3

1.9 - Figure 1. The alignment of open-close trajectories is really hard to see. We have improved the image in the resubmitted manuscript.

1.10 - Figure 2. In panel A, it is the same structure.

We are very thankful for pointing out this mistake. It has been corrected.

1.11 - Some of the arrow representations are too dense and it is difficult to grasp the overall motion. Could you reduce the number of arrows to improve the figures?

The arrow representations of the PCs have been improved in all figures.

Reviewer #2 (Remarks to the Author):

We thank the reviewer for insightful comments and constructive criticism that, in our opinion, has helped to significantly improve our manuscript and achieve more solid conclusions.

A. The overall aim of this work is to predict and validate protein structures that lie between the known endpoints of conformational transitions. To this end, the authors have performed Langevin dynamics and compared the trajectories from their simulations against known structures in between the end points.

2.1 - The figure showing this comparison has been placed in the supplemental material, and yet this is one of the most important outcomes.

We understand the referee was referring to the Figure S1 in the original manuscript, which showed the rMSD and PC1-2 distance evolution along the trajectories. **We have realized rMSD is a more general measure for structural distance between conformers than PC1-2 distance** (i.e., it does not need previous ensemble analysis) and thus should be included in the main material to allow for direct comparison. Accordingly, in the resubmitted manuscript we have included these graphs in the main figures and have emphasized the difference in using PC1-2 distances versus rMSD in the *Methods* sections.

See: Discussion and Methods, Figs. 2-5 and Fig.7 (panel D in all).

Interestingly they have performed simulations in the two different directions between the endpoints and importantly observe different pathways in the two directions.

2.2 - The transitions from closed to open forms have always been more difficult to achieve with such elastic models so this results showing transitions in that direction are important.

A more thorough discussion of trajectory asymmetry and its relation to the free energy landscapes has been added to the paper (*see point 2.6 and 2.8 below*); we must note here however that **the difference in the trajectories starting from closed or open conformers is very small, specially when the structures can be hardly defined as “open” or “closed” (5-NTase, GLIC) in terms of the R_g** (see Table 0 at the end of this letter). We also noticed that the nomenclature used for GLIC, for example, was misleading (the so called “open-pore” i.e. conducting channel has a “closed” ECD). **In order to avoid confusions we have included the R_g for all structures and a better description of their features.**

See: Supplementary Table 1

B. The novelty of the work lies primarily in the Langevin simulations, and in using them to perform simulations in two different directions.

2.3- But they have not been sufficiently described, and it remains unclear whether the parameters involved have been sufficiently tested and validated.

We thank the reviewer for its appreciation of the Langevin simulations, and for bringing this lack of clarity in method description and previous references to our attention. We have included a **detailed description of the choice for the different eBDIMS parameters in the *Methods* section**, and also included figures showing the effect of the free variables (see *point 2.5* below).

See: *Methods, Fig.S10*.

There are a number of important points, however, that the authors have not addressed in the present manuscript:

2.4 - How reliable are the PC's? *There is the important issue of whether the set of available structures is sufficient for developing reliable PC's. Eleven and sixteen structures for two of the structure sets used here are very small numbers. In our past experience such small sets can yield significantly distorted views of the conformational space. A recent paper in J Chem Phys investigated the convergence of PCs using different numbers of structures, which should be done here.*

We totally agree with the referee that the ensembles are reduced for some of the proteins, but in our opinion, **they truly represent the conformational space as can be seen by their correlations with heuristic reaction coordinates** defined in the literature. As discussed above with *Referee #1 (point 1.2)*, one can find in the Protein Data Bank ensembles containing a great number of structures, but often they cannot yield robust PCs because they do not have clear on-pathway intermediates.

During the manual curation process in search for transition intermediates, it became apparent to us that **the robustness of PCA and its clustering of functional states is not as dependent on the number of structures as on their sampling of distinct conformations distributed along a path**. This is especially relevant for large-scale changes such as those in the Weiss and Levitt's benchmark¹ studied here, which in spite of the small number of structures includes clear transition intermediates. For that reason, we included structures with repaired gaps as long as they fall in the same clusters defined by PCA of intact structures. The **importance of the quality or width of the sampling rather than the quantity of structures is clear considering two extreme examples:**

- As discussed with Referee #1, we have collected a high number of X-ray and NMR structures for BPTI (near 200 structures), but they render ill-defined PCs that are extremely sensible to ensemble composition; here, C-alpha PCA is of no use to cluster functional states and benchmark transition pathways (see *Fig.B* at the end of this letter)

- The ensemble for **SERCA**, which re-produces the PCA major axes with only **three-four structures** belonging to different clusters compared to 65 near-intact structures (see Supplementary Table 2).

To illustrate this point we **report the dot products** (Supplementary Table 2) **between the reference intact ensembles and “reduced” ensembles with a minimal number of structures**. As can be seen, **the robustness of the first PCs is extremely high** for these ensembles that sample large conformational changes with true intermediates trapped along a transition, and provides a similar clustering of the structures (Supplementary Figure 1) and variance distribution. In fact, the less robust ensemble is that of GLIC (which has the lowest rMSD) in spite of its rather large number of structures (46). With **just three structures (end-states plus an intermediate), all the ensembles render PCs that overlap >70%** with those from the full ensemble. This supports the idea that **few but widely distributed structures in the conformational space render more robust PCs than a large number of redundant structures sampling similar conformations**.

See: Methods, Fig.S1 and Supplementary Table 2

2.5 - ...Langevin simulations have not been sufficiently described, and it remains unclear whether the parameters involved have been sufficiently tested and validated. The Langevin simulation parameters have not been sufficiently described. Replicas of the simulations have not been performed to validate the trajectories.

We appreciate that the referee points out this lack of detail in our description of the method parameterization and validation. As mentioned above (point 1.4), eBDIMS was developed from our previous algorithms for unbiased **Brownian Dynamics** (see 7,10,11) and the ED-ENM force-field for NMA (see 11,12), which were both carefully parameterized against our MoDEL database^{13,14} (<http://mmb.pcb.ub.es/MoDEL/>) of state-of-the-art MD simulations for the main protein metafolds, as well as against experimental data from X-ray crystallography and NMR. The coupling parameters for Langevin simulations as well as the ED-ENM Hamiltonian were thoroughly optimized (point 3.4) to reproduce atomistic MD. For the topology-based ED-ENM potential function, force constants were fitted multi-parametrically to reproduce the sampling by state-of-the-art standard force fields (AMBER, GROMOS, OPLS and CHARMM) at the C-alpha carbon level. The Langevin simulation thermostat, based on the fluctuation-dissipation theorem, and implicit solvent representation given by the friction term, were also fitted to MD simulations following a similar scheme.

We want to emphasize that the force-constants used here (ED-ENM) and the BD parameters were **refined to reproduce subtle anharmonic features of atomistic simulations that are not well captured by standard ENM methods**¹⁵, such as the variance distribution of the eigenvalues given by Essential Dynamics (ED) analysis or the coupling forces between C-alpha carbons. Other authors have applied our C-alpha force field for advanced applications such as evaluating transition energies to

score “ab initio” CASP predictions because of its agreement with MD¹⁶; we also used both algorithms in a recent study of beta-sheet correlated motions¹⁷ published in this journal.

As discussed also with Referee #1 and #3, **using the default optimal parameters the variability of the trajectories is minimal and they tend to converge even if started from different structures along a path (Fig. S10A)**. However, by changing the number of unbiased steps, k , it is possible to increase sampling width (Fig. S10C). We have run several replicas for some of the examples to illustrate the effect of changing these free variables (random seed and k).

See: *Methods, Fig.S10*

E. The energetics has been completely ignored, with only geometry and entropy being considered.

2.6 - If energies were considered the trajectories might be significantly distorted.

As mentioned above, the **Brownian Dynamics simulation and Elastic Network force field were carefully calibrated using atomistic simulations**. Thus, the energetics was implicitly considered as long as we overlap with MD in 60-80% in directions, amplitudes of motions and forces acting between C-alpha carbons; note that this value is similar to that obtained when comparing standard MD force-fields among them (as shown in our previous works^{7,12}). Furthermore, the fluctuation-dissipation relation assures thermal energy keeps stable throughout the BD simulation. Nevertheless, one must keep in mind that we are using a simplified Hamiltonian based on the minimum frustration principle, i.e. the pathway collected is that leading from start to end structures with the minimum frustration of native contacts. The method is **by definition coarse-grained and thus not aimed to provide an accurate evaluation of the free energies** of transitions (a challenging task even for MD). However, considering that

- i) **The sampled routes totally converge with those from *Climber*, based on the fully-atomistic Molecular Mechanics Force Field ENCAD,**
- ii) **They are populated by experimental intermediates,** which presumably correspond to meta-stable structures

Together indicates that the explored pathways, although based on a topological/geometry-based potential, do correspond to feasible low-energy routes in the conformational space. Precisely, the **PCA framework is intended to provide immediate validation of explored pathways taking advantage of the experimental information** available. Assuming that an intermediate crystal structure represents a metastable state, one must conclude that the routes approaching them are energetically possible, a notion supported by MD simulations (*See below, point 2.8*).

2.7 - The authors could also have utilized atomic elastic models, which would have a different set of potential functions from the coarse-grained cases.

Certainly, our method could be easily adapted to all-atom or finer-grained representations; in fact, **we have previously developed atomistic approaches for transition path-sampling using discrete Molecular Dynamics** (see our previous works^{5,6}). In our experience, **the increase in the frustration energy originated from raising the number of contacts makes calculations less efficient without increasing pathway accuracy**. A number of works have compared geometric/topology-based paths with those from atomistic simulations using TMD, umbrella sampling, etc, and have demonstrated that *“finding an all-atom pathway is primarily a problem of geometry, and that a detailed force field in this case constitutes an unnecessary extra layer of detail”*¹⁸. Similarly, it is well known that **all-atom and C-alpha ENM provide nearly equivalent representations of protein equilibrium dynamics** (demonstrated in the seminal paper by Tirion¹⁹), and in our experience, coarse-grained ENM is better suited than atomistic NMA to track large and collective motions, as seen in structure pairs like those studied here. In our opinion, **considering that the trajectories sampled by eBDIMS and Climber are virtually identical supports that path sampling is a “shape” (i.e. topological) problem independent of atomistic details**, and that introducing side-chain modeling would contribute marginally to improve the method.

2.8 - This omission of energies raises serious questions about the reliability of the results. In general there must be some energy-entropy compensation along the trajectories since the closed forms would be expected to have lower energies and the open forms higher entropies. In fact these differences may account for why the trajectories differ in the two directions... *Adding some evaluation of the energies is critical to discern whether these trajectories lie in two separate free energy troughs or how these particular trajectories relate to the details of the free energies.*

We thank the referee for bringing this important point to our attention. Although we were aware of pathway asymmetry in the PC1-2 space, taking into account PCs relative variance, it is in fact low for most of the examples studied; in the resubmitted manuscript, a pathway asymmetry score has been introduced to quantify such divergences in the PC1-2 space (See *Methods* and *Fig.S4*). In principle, an accurate evaluation of the free energies of the transitions was beyond the scope of the present work, and in our opinion, of coarse-grained methods. We **relied instead on the comparison with crystallographic intermediates** (which presumably correspond to transitional states trapped along the free-energy troughs connecting end-conformations) **as immediate estimation of a pathway in energetic terms**. The total **convergence with the results from the ENCAD²⁰ atomistic force-field used by Climber also supports that the explored pathways are energetically correct**.

Although a role for pathway asymmetry has been suggested for some proteins (see for example²¹), it can also arise from introducing a bias, which breaks the detailed balance condition for equilibrium. Here, in spite of the entropy-energy compensation of the Langevin thermostat, to drive a transition we use a Maxwell's evil introducing information in the system, which means that *de facto* entropy

changes as the trajectory advances and Boltzman's sampling is biased; similarly happens with the pulling algorithm used by *Climber*. In principle, **we considered pathway asymmetries to be non-significant for most proteins when considering the variance**; further evaluation of heuristic variables suggested that, in general, the paths were fairly reversible. However, as can be seen for the example of RNaseIII, **crystallographic intermediates can appear in both directions**. Considering that these structures represent landmarks along minimal energy paths, pointed that the observed asymmetries could be meaningful in some cases.

To further elaborate on this interesting question (i.e. the relation of pathway asymmetries with lowest energy troughs), we have **compared eBDIMS with the atomistic free-energy landscapes from multi-microsecond MD** (also following the suggestion by Referee #1). Apparently, the biased trajectories from the closed state (entropy-driven, as the referee points out) and the open state (energy-driven) provide two alternative and valid solutions to the path. Our comparisons hint that **the forward/reverse pathways** predicted by eBDIMS and *Climber* **delimit, not a linear route, but an area that corresponds to the lowest energy troughs** connecting the end-states sampled by MD, and that the **crystallographic intermediates tend to populate the boundaries of these regions** rather than the explored minima that lie within. Only for RNaseIII (with the highest asymmetry) both MD and the distribution of crystal structures suggest potentially different forward/reverse paths (*Fig.S5C*)

See: Results - Sampling of intermediate states in atomistic free energy landscapes, and related Figures (Fig.9), and Discussion; Methods and Fig.S4

F. Summary of the problems with the paper.

- *The PC's require some validation to justify the use of such small numbers of structures. Addressed above (Supplementary Table 2).*
- *The work needs to describe in more detail the simulations and how the parameters for them were chosen. Corrected (Updated Methods)*
- *Replicas of the simulations need to be performed to observe their variabilities.- Addressed above (Supplementary Fig. 10)*
- *Adding some evaluation of the energies is also critical to discern whether these trajectories lie in two separate free energy troughs or how these particular trajectories relate to the details of the free energies.- Addressed above (Fig.9)*

G. There are a number of additional references that could have been cited.

We have included a number of additional references, but **could not find the one suggested by the referee about PC robustness on *J.Chem.Phys.*** We will be thankful to check and include it if he/she can direct us to this citation/s.

H. The paper is clearly written.

Reviewer #3 (Remarks to the Author):

The work by Orellana et al. contains two attractive approaches: first, for classifying multiple structures of proteins (or protein complexes) within a single conformational landscape; second, for determining probable transition pathways between distinct structural groups. As more and more proteins are solved in different conformational states, these approaches will find many applications and should help to understand the functional mechanisms of the proteins in question. The methods that have been developed are clearly described and their application to four well-studied cases is nicely documented and gives rise to some interesting results.

We thank the reviewer for the encouragement and support and are pleased that (s)he finds the study of interest. We would like to note that a further example has been added to the main material (*Fig.5-6*), and we are preparing a curated database (some examples at the end of this letter, *Fig.A*) that will include cryo-EM and NMR structures, to show the generality of the method for rationalizing and completing experimental information on the conformational landscape.

My main queries involve the automated application of these techniques to structural data, particularly when little or nothing is known about the function of the proteins studied:

3.1 - Why is it necessary to use an identified apo structure as the reference in the PCA analysis? I can understand this choice in the case of elastic network models where the contact network of "closed" structures makes closed-to-open transitions more difficult than the reverse, but what would change if the PCA analysis used another structure as the reference? In the future, it is probable that cases will arise where the apo/holo distinction is not known a priori.

The referee makes a very good point, and we realize that we have not clearly stated the reasons for reference selection in the manuscript. As a matter of fact, **the reference structure has no influence on PCA clustering; any structure can be used for that purpose and the only change is a displacement of the origin of coordinates.** Typically, the average of the ensemble is taken as reference for alignment and projection, but these coordinates usually do not correspond to a real structure but rather to the middle point of the clusters (which can be an unpopulated area). In order to **make it easier to interpret the PCs as deformations from a real structure, we found it more natural to use the inactive/apo/resting states as reference** rather than a geometric average. Apart from that, since we are running eBDIMS in both directions, any of the end-states (bound or unbound, closed or open, apo or holo or any other pair) could work for comparisons. We have improved the description of this issue in the *Methods* section.

See: *Methods*

3.2 - How robust are the PC's to the removal of one or more structures within a given structural group?

As discussed above with Referee #2 (*see point 2.4*), we have addressed this issue by computing the PCs for a minimal number of structures and then calculating their overlap with the full ensembles (*Supplementary Table 2*). For collective conformational changes, the two major PCs are **>70% identical for most proteins with just three structures (one from each end-state plus an intermediate)**. The variance distribution and clustering is also

See: *Fig.S1. and Supplementary Table 2*

3.3 - I did not understand the definition of the progress variable gamma. What is meant by the "internal distances" d_{ij} and how do they direct the dynamics to the target structure?

Thanks for pointing out this lack of clarity in the algorithm description. We have improved the text describing eBDIMS and completely rephrased the description of the progress variable in the *Methods* section. The **progress variable gamma measures the differences in pairwise distances between residues (d_{ij}) in the starting and target structures**, so that intermediate conformations that reduce this difference are selected every certain number of iterations (k) (*Fig.S9*). As we now explain with detail, the d_{ij} differences do not need to be known for all the residues, and the method can actually work with a minimal set of distance restraints or even when such restraints are lacking for large parts of the target structure (see practical example *Fig. S6A*); the algorithm runs faster as more information from the target is introduced. Speed can be increased also by reducing the number of unbiased steps k , but this does affect the sampled pathways, which tend to progress too fast to properly sample the intermediates (similarly observed for Climber¹); reducing the value of k has the opposite effect, allowing the algorithm to wander and generate more random, MD-like trajectories (*Fig. S10C*).

See: *Methods and Fig.S6, S9 and S10*

3.4 - Do any modifications have to be made to the ebDIMS procedure for treating different protein structures? For example, the number of unbiased steps "k".

As we highlight in the revised manuscript, **the default values of both the BD simulation (friction coefficient, temperature, etc) and the ENM potential (force constant definition) were thoroughly parameterized based on a database of MD simulations** and are thus optimal to treat any protein (*see also response to Referee #2, point 2.5*). However, for very large proteins or challenging conformational changes it can be useful to increase the biasing frequency (k) in order to speed up calculations keeping in mind that this tends to make trajectories closer to a straight interpolation.

See: *Methods*

Fig. A. Further examples of multi-state ensembles with possible on-pathway intermediates (in red). For the HCV-helicase (A), PC1-2 projections distinguishes the open structures crystallized together with the protease domain (1cu1 and others) from those in which it has been removed (3kqu-like), detecting intermediates already described in the literature (3o8c, 3kqh or 3kqk) and others not characterized as such (mutant 2f55). Similarly happens with Calmodulin, the Catabolite Repressor protein or Importin, with structures that appear as possible transient states between end-conformations that have been stabilized upon binding to different ligands.

Fig.B. Lack of robustness of BPTI ensembles for C-alpha PCA. The PCs of BPTI are completely different when including/excluding NMR structures or to the removal of a few structures of each group; furthermore, clustering is not related to functional state (bound/unbound) or any other clear variable (i.e. experimental conditions etc). Note that N- and C-termini were removed (2 residues each) to reduce noise from their fluctuations.

Table 0. Iteration Times with ebDIMS (rMSD to target in parenthesis, last column)

Name	State1	State2	RMSD _t	Transitions	ebDIMS Iterations
RBP	1BA2(A)	2DRI	6.2	Forward (closing)	26914 (1.0 Å)
	Open (21.4)	Closed (19.8)		Reverse (opening)	26062 (1.0 Å) faster
	Unbound	Ribose-bound			
5'-NTase	1OID	1HPU	9.3	Forward (opening)	44051 (2.0 Å)
	"Closed" (21.5)	"Open" (21.6)		Reverse (closing)	43884 (2.0 Å) faster
	Unbound	Nucleotide-bound			
RNaseIII	1YYO	1YYW	17.8	Forward (opening)	91002 (4.0 Å) faster
	Closed (24.5)	Open (26.6)		Reverse (closing)	110632 (4.0 Å)
	dsRNA-bound	dsRNA-bound			
SERCA	2C9M	1T5S	14.16	Forward (closing)	144483 (3.0 Å)
	Open headpiece (38.6)	Closed headpiece (37.7)		Reverse (opening)	112193 (3.0 Å) faster
	Ca ²⁺ -bound	Ca ²⁺ /Nucleotide-bound			
GLIC	4NPQ	4HIF	2.6	Forward (closing ECD)	27032 (1.0 Å)
	Open ECD (37.6)	Closed ECD (37.2)		Reverse (opening ECD)	26623 (1.0 Å) faster
	Closed pore	Open pore			
	Resting (pH=7.5)	Conducting (pH=4)			

References

1. Weiss, D. R. & Levitt, M. Can Morphing Methods Predict Intermediate Structures? *J. Mol. Biol.* **385**, 665–674 (2009).
2. Smolin, N. & Robia, S. L. A structural mechanism for calcium transporter headpiece closure. *J. Phys. Chem. B* **119**, 1407–1415 (2015).
3. Gur, M., Zomot, E. & Bahar, I. Global motions exhibited by proteins in micro- to milliseconds simulations concur with anisotropic network model predictions. *J. Chem. Phys.* **139**, (2013).
4. Shaw, D. E. *et al.* Atomic-Level Characterization of the Structural Dynamics of Proteins. *Sci.* **330**, 341–346 (2010).
5. Sfriso, P., Hospital, A., Emperador, A. & Orozco, M. Exploration of conformational transition pathways from coarse-grained simulations. *Bioinformatics* **29**, 1980–1986 (2013).
6. Sfriso, P., Emperador, A., Orellana, L., Hospital, A. & Orozco, M. Finding Conformational Transition Pathways from Discrete 2 Molecular Dynamics Simulations 1. (2012).
7. Emperador, A., Carrillo, O., Rueda, M. & Orozco, M. Exploring the suitability of coarse-grained techniques for the representation of protein dynamics. *Biophys. J.* **95**, 2127–2138 (2008).
8. Carrillo, O., Laughton, C. A. & Orozco, M. Fast Atomistic Molecular Dynamics Simulations from Essential Dynamics Samplings. *J. Chem. Theory Comput.* **8**, 792–799 (2012).
9. Chaudhuri, R., Carrillo, O., Laughton, C. A. & Orozco, M. Application of Drug-Perturbed Essential Dynamics/Molecular Dynamics (ED/MD) to Virtual Screening and Rational Drug Design. *J. Chem. Theory Comput.* **8**, 2204–2214 (2012).
10. Orozco, M. *et al.* Coarse-grained Representation of Protein Flexibility. Foundations, Successes, and Shortcomings. *Adv. Protein Chem. Struct. Biol.* **85**, 183–215 (2011).
11. Camps, J. *et al.* FlexServ: an integrated tool for the analysis of protein flexibility. *Bioinformatics* **25**, 1709–10 (2009).
12. Orellana, L. *et al.* Approaching elastic network models to molecular dynamics flexibility. *J. Chem. Theory Comput.* **6**, 2910–2923 (2010).
13. Rueda, M. *et al.* A consensus view of protein dynamics. *Proc. Natl. Acad. Sci. U. S. A.* **104**, 796–801 (2007).
14. Meyer, T. *et al.* MoDEL (Molecular Dynamics Extended Library): a database of atomistic molecular dynamics trajectories. *Structure* **18**, 1399–1409 (2010).
15. Rueda, M., Chacón, P. & Orozco, M. Thorough validation of protein normal mode analysis: a comparative study with essential dynamics. *Structure* **15**, 565–75 (2007).
16. Perez, A., Yang, Z., Bahar, I., Dill, K. A. & MacCallum, J. L. FlexE: Using Elastic Network Models to Compare Models of Protein Structure. *J. Chem. Theory Comput.* **8**, 3985–3991 (2012).
17. Fenwick, R. B., Orellana, L., Esteban-Martín, S., Orozco, M. & Salvatella, X. Correlated motions are a fundamental property of β -sheets. *Nat. Commun.* **5**, 4070 (2014).
18. Farrell, D. W., Lei, M. & Thorpe, M. F. Comparison of pathways from the geometric targeting method and targeted molecular dynamics in nitrogen regulatory protein C. *Phys. Biol.* **8**, 26017 (2011).
19. Tirion, M. Large Amplitude Elastic Motions in Proteins from a Single-Parameter, Atomic Analysis. *Phys. Rev. Lett.* **77**, 1905–1908 (1996).
20. Levitt, M., Hirshberg, M., Sharon, R. & Daggett, V. Potential energy function and parameters for simulations of the molecular dynamics of proteins and nucleic acids in solution. *Comput. Phys. Commun.* **91**, 215–231 (1995).
21. Seyler, S. L. & Beckstein, O. Sampling large conformational transitions: adenylate kinase as a testing ground. *Mol. Simul.* **40**, 1–23 (2014).

Reviewers' Comments:

Reviewer #1 (Remarks to the Author)

The authors addressed many of the suggestions raised by this referee. However, the work done has been not fully incorporated in the present version, specifically:

R 1.1 Since we all agreed in the limitation of the dataset, please include the examples of Fig. A at least as supplementary material. Definitively, the inclusion of more test cases will add more strength to the validation of your approach.

R 1.2A In [3] the ANM collective modes define the pathways between the crystal structure and several well defined PCA sub-states sampled in Anton simulations. My suggestion was in the direction of reproducing these results and to explore how eBSIMS performs in transitions observed by very long MD. I do not think coarseness is an issue here, in fact ANM is even more coarse than eBDIMS. However, if eBDIMS is not suited to handle this collective but small changes this should be comment in detail in the text.

R 1.2 B The free-energy landscapes plots are noteworthy results. It is nice to see how well the pathways can delimited the low-energy routes. I also agree that path sampling is mainly a shape problem. For this reason, here, I also miss the comparison with NMA approaches that are the paradigm of "shape" methods. By comparing the corresponding figures of RBP, SERCA and GLIC, it is clear that the non-linear NMA method, iMODs, shown similar performance. Also iEMN seems to walk by the lowest energy areas. Please, illustrate and comment this observation. Since you have the 5-NTase MD simulation (Fig. S5), I am missing the corresponding energy landscape plot. Also in this figure and for consistency, the initial SERCA structures of figure 5A and figure 9B should be the same.

R1.3 The comparison with the state of the art methods is mandatory to illustrate the goodness and limitations of the proposed approach. To this respect NMAD-ref and MinActpath are valuable but a bit obsoleted (e.g. the servers are not updated since 2006-7) and it will be more fair the inclusion of more recent approaches. Although, the scope of the paper is not a detailed comparison, few comments about the relative performance in each case should be included. Right now, the authors just added the pathways plots (panel C), but there are not practically any comments in the text. This should be corrected. By the way, transition progress plot could be easily improved by using a dash-line and continue line for forward and reverse pathways, respectively. Also, in figures 3 and 4 iMODs seems to be under-sampled.

The word "instability" is too ambiguous and it is clear that different methods have different behaviors and "instabilities". NOMAD-ref and iEMN clearly fail to reach the final structure in the RNAase case. By the contrary, NMSIMS except in the RNAase case have an unexpected behavior fluctuating around near final positions. This must be further investigated and commented. For example, I run RNAase case in the iENM server and I notice that it got trapped in local minima in where one of the binding domains collided. I also try to run the same case with NOMAD-ref, but I always obtained quite distorted models. Again, this should be investigated and commented. In sup. Figure S3 you are using a failed case (panel B) to illustrate the performance of NOMAD-ref that is likely to be an artifact. Also, in this figure more recent methods were ignored.

It will be useful include a table with the minimum RMSDs of the different methods respect to the end conformation, as well as, the minimal distance to the representative intermediate structures. Ideally, you could include the transition frame plots (as panel D, fig 2-5) for all the tested methods.

All these programs have different input parameters (iterations, cutoffs, number of modes, etc.) please specify them to facilitate the reproducibility of your results.

The authors should stress more clearly the limitations of the propose approach, case by case:

-RBP. The reverse pathway of eBDIMS (also at a least extent climber) clearly stands out of the lowest energy regions. Apparently, this is a type of transition in where NMA methods have similar, if not better, results than climber and eBDIMS.

-5'-NTase. The reverse Climber and eBDIMS transitions followed a path not observed in MD.

-RNaseII. The closest conformation of eBDIMS to the catalytic, functional state is around 10 Å.

-All the method seems to be capable to sample the GLIC and SERCA transitions, but will be interesting to see the rMSDs differences.

Reviewer #2 (Remarks to the Author)

The authors have made an extraordinary effort to respond to the reviews, and have made the paper significantly more interesting.

The missing reference that I should have provided is:

Sankar K, Liu J, Wang Y, Jernigan RL. Distributions of experimental protein structures on coarse-grained free energy landscapes. *J Chem Phys.* 2015 28;143(24):243153.

One particularly interesting transition was one studied by X-ray where the changes in the intermediates were more significant than the differences between the end point structures. This could be the subject of a future work. See: Schotte F, Lim M, Jackson TA, Smirnov AV, Soman J, Olson JS, Phillips GN Jr, Wulff M, Anfinrud PA. Watching a protein as it functions with 150-ps time-resolved x-ray crystallography. *Science.* 2003 20;300(5627):1944-7.

Reviewer #3 (Remarks to the Author)

The authors have addressed all my queries. The addition of a new sample protein and also more detail in describing the methodology certainly helps to improve the impact of this work. In my opinion, the revised manuscript satisfies all the criteria for publication.

Reviewer #1 (Remarks to the Author):

The authors addressed many of the suggestions raised by this referee. However, the work done has been not fully incorporated in the present version.

Thanks to the referee for the additional suggestions to improve the manuscript. We have tried to address the remaining points taking into account the strict limitations on article length – which preclude more elaboration of some interesting topics such as detailed comparisons between methods, etc – and on time for submission (3 weeks).

R 1.1 Since we all agreed in the limitation of the dataset, please include the examples of Fig. A at least as supplementary material. Definitively, the inclusion of more test cases will add more strength to the validation of your approach.

We have included as suggested these cases in the supplementary material (Supplementary Figure 2 and Supplementary Table 3) **just to show the detection of possible on-pathway intermediates defined by eBDIMS and X-ray PCA**; however, as mentioned on the rebuttal letter **they require further investigation on a case by case basis to determine their possible significance as done for the presented model proteins** (literature search, MD simulations, etc), and thus we do not include them in the main material.

R 1.2A In [3] the ANM collective modes define the pathways between the crystal structure and several well-defined PCA sub-states sampled in Anton simulations. My suggestion was in the direction of reproducing these results and to explore how eBDIMS performs in transitions observed by very long MD. I do not think coarseness is an issue here, in fact ANM is even more coarse than eBDIMS. However, if eBDIMS is not suited to handle this collective but small changes this should be comment in detail in the text.

We feel here that there is some misunderstanding regarding the difficulties in dealing with the example suggested by the referee.

We must note that **the default potential used by eBDIMS is actually an elastic network developed for ANM (See paper¹), and thus the coarse-graining is identical (the only difference is the friction/solvent term missing in NMA)**. As the referee indicates, ANM can deal easily with conformational changes, and therefore, eBDIMS can as well. In fact, small transitions such as these can be perfectly handled by eBDIMS, even at finer grained resolutions. **The problem for the discussed example is not due to eBDIMS, the ANM potential or the level of coarse-graining, but to the experimental ensemble itself**, that prevents the calculation of robust PCs at least at the C-alpha carbon level.

Contrary to the other examples presented, the PCA results for BPTI change dramatically

depending on the structures chosen (for example, when NMR structures are included/excluded, etc), and there is no apparent pathway between end-states well defined experimentally. Therefore, **we cannot compare ANM, ebDIMS or MD in a robust and independent PC-space to validate their relative performance with experimental data: the BPTI ensemble PCs do not cluster neatly the solved structures** (for example, separating bound/unbound, etc), and this is a problem related to the magnitude of the conformational change, which is extremely small at the C-alpha carbon level – there are not clear collective motions such as hinge, twist, etc but rather local backbone fluctuations. We already **discussed this issue of PCA robustness in the previous rebuttal letter and incorporated in the manuscript.**

R 1.2 B The free energy landscapes plots are noteworthy results. It is nice to see how well the pathways can delimit the low-energy routes. I also agree that path sampling is mainly a shape problem. For this reason, here, I also miss the comparison with NMA approaches that are the paradigm of "shape" methods. By comparing the corresponding figures of RBP, SERCA and GLIC, it is clear that the non-linear NMA method, iMODs, shows similar performance. Also iEMN seems to walk by the lowest energy areas. Please, illustrate and comment this observation.

We have commented with more detail as suggested the differences between methods (**Supplementary Discussion**). The referee is right that all methods are capable of finding one of the lowest energy "boundaries" (the most populated one in MD) but only for examples with low pathway asymmetry. However, **for RNaseIII, in which both pathways are clearly distinct, most classical approaches fail to provide a stable path,** and only MinActionPath, which also uses Langevin dynamics, provides a smooth pathway, although only in one direction.

Since you have the 5-NTase MD simulations (Fig. S5), I am missing the corresponding energy landscape plot.

In principle, we did not aim to get a FEL for all the proteins because not all examples have been studied previously and besides that, they contain ligands of different complexity (an entire dsRNA for RNaseIII). We preferred to focus on those cases which have been well characterized computationally and experimentally and are known to transition without the ligand: RBP, SERCA and GLIC. **Regarding our simulations of 5'NTase (never studied computationally), we can only conclude that partial closing seems spontaneous** in the absence of ligand and samples most of the area defined by ebDIMS/Climber as seen in our MD simulations. However, **these simulations from unbound intermediates and end-states are not fully transitioning (the area between the two basins, 1HPU and 1O18 is undersampled) and thus do not overlap (as also happens for RNaseIII), preventing the calculation of a meaningful FEL.**

Also in this figure and for consistency, the initial SERCA structures of figure 5A and figure 9B should be the same.

For all examples, we preferred to use already published data so that the readers can find a detailed analysis of the corresponding MD simulations in the literature. For SERCA, the only reported simulations that transition spontaneously between the open and the closed state start from the 1SU4 structure. **This structure and the one we used initially as seed (2C9M) are known to belong exactly to the same functional state and as can be seen, they populate the same area in the PC1-2 subspace**, which corresponds to the E1-2Ca²⁺ state. This Calcium-bound open headpiece conformation is known to be extremely flexible, hence the wide distribution of this cluster, but as discussed in the Robia paper, **the open Calcium-bound SERCA (which samples all the area of 1SU4 and 2C9M structures) is known to transition to the closed intermediates as determined experimentally**. Running eBDIMS from any of the open structures (1SU4 or 2C9M chains A or B), the pathways visit the intermediates suggested by the PC analysis. Note that **we already added the forward/reverse transitions from 1SU4 in the FEL for consistency**, as suggested.

R1.3 The comparison with the state of the art methods is mandatory to illustrate the goodness and limitations of the proposed approach. To this respect NOMAD-ref and MinActionPath are valuable but a bit obsoleted (e.g. the servers are not updated since 2006-7) and it will be more fair the inclusion of more recent approaches. Although, the scope of the paper is not a detailed comparison, few comments about the relative performance in each case should be included. Right now, the authors just added the pathways plots (panel C), but **there are not practically any comments in the text. This should be corrected**.

We do not agree fully with the referee on this point: in our experience, algorithm quality is not related with its novelty (at least as seen from projections onto experimental PCs) but rather to the physics behind. As can be seen, in spite of being an “old” method, **MinActionPath (2007) is not obsolete but actually the best ANM-based linear algorithm, showing smooth trajectories close to Climber and ebDIMS for all the examples studied. Notably, it also uses a Langevin simulation**.

On the contrary, a more recent and complex approach such as NMSIMS (2012), which includes three levels of structure modeling, or iMODS (2014), based on an intricate internal coordinates NMA interpolation, show rather “jagged” trajectories in the PC1-2 subspace for the challenging RNaseIII transition.

To our knowledge, we have included all the most recently developed methods (iMODS, NMSIMS) as suggested by the referee. **The only exception is AMPathway (2014), and the reason is that the server only returned results for GLIC**. The projection of this transition shows a similar behavior as the other pure ANM-based methods. We also

excluded our own recent approaches based on dMD^{2,3} because they cannot deal with multichain structures.

In summary, method exclusion or inclusion has been solely based on the capacity of servers and programs to deal with the examples examined, and it is clear that their relative performance is only related to their physics, with old methods such as MinActionPath working better than more complex/recent ones such as NMSIMS (which have other advantages, such as dealing with RNA, or being extremely fast as iMODS).

By the way, transition progress plot could be easily improved by using a dash-line and continue line for forward and reverse pathways, respectively.

In our opinion, this way **the plots where other methods are included would appear even more crowded**; note that we added “forward” and “reverse labels” to the eBDIMS plots, and the gray gradient also allows to identify transition direction.

Also, in figures 3 and 4 iMODS seems to be under-sampled.

The data for **iMODS was obtained using defaults** from the server, for each one of the cases studied. The default is set to 1A for each step, so that for a smaller the transition, there are less steps (See updated Supplementary Methods).

The word "instability" is too ambiguous and it is clear that different methods have different behaviors and "instabilities". NOMAD-ref and iEMN clearly fail to reach the final structure in the RNAase case. By the contrary, NMSIMS except in the RNAase case have an unexpected behavior fluctuating around near final positions. This must be further investigated and commented. For example, I run RNAase case in the iENM server and I notice that it got trapped in local minima where one of the binding domains collided. I also try to run the same case with NOMAD-ref, but I always obtained quite distorted models. Again, this should be investigated and commented.

As we pointed throughout response letter, we did not aim to rate methods or analyze the weaknesses of each algorithm, just to use experimental data as a framework to evaluate how they sample transitions. **By instabilities we refer to the “jaggedness” of the trajectories projected in the PC1-2 space, and not to their causes, which can be clearly related to the differences in the algorithms and should require a detailed examination of the codes (not accessible in many cases) clearly beyond the scope of this work.**

However, **it seemed clear that a very common and understandable problem for ANM methods is to pick up the right biasing mode at each step**, and we illustrate that with NOMAD-Ref because it was developed by ourselves and clearly fails in several examples.

In sup. Figure S3 you are using a failed case (panel B) to illustrate the performance of

NOMAD-ref that is likely to be an artifact. Also, in this figure more recent methods were ignored.

The goal of Fig.S3 was to illustrate the reason for NOMAD-Ref failing to find a stable path, not to show its performance that is clearly bad in the cases presented. **There is no point in including recent methods that perform well (they are already included in the main material), since we aimed to show why an ANM method such as NOMAD-Ref fails when selecting the wrong normal modes to deform a structure.**

It will be useful include a table with the minimum RMSDs of the different methods respect to the end conformation, as well as, the minimal distance to the representative intermediate structures. Ideally, you could include the transition frame plots (as panel D, fig 2-5) for all the tested methods.

As suggested, we have included RMSD and PC-distance comparisons for all the examples (Supplementary Table 5); however, **note that the distance to crystal intermediates has little relation with methods defining smoothly the wider area sampled by MD, and thus is a poor indicator of method performance** (jagged trajectories can reach very low rRMSDs to intermediates but are not physically feasible).

All these programs have different input parameters (iterations, cutoffs, number of modes, etc.) please **specify them to facilitate the reproducibility of your results.**

Default values set in the webserver or the provided executables (Climber) were used for all programs to assure reproducibility. We include links to all webserver for all the methods (Supplementary Table 4) and report the values of the main parameters according to original papers (Supplementary Methods). Note that in many webserver there is no option to select any parameter at all (iENM, for example).

The authors should stress more clearly the limitations of the proposed approach, case by case:

-RBP. The reverse pathway of eBDIMS (also at a least extent climber) clearly stands out of the lowest energy regions. Apparently, this is a type of transition in where NMA methods have similar, if not better, results than climber and eBDIMS.

We do not agree with the referee interpretation. Linear methods only show one of the boundaries of the region sampled by MD, while Climber and eBDIMS delimit both of them. The reverse pathway from the closed 2DRI structure sampled by eBDIMS and Climber is covered by MD in most of its extension, and approaches significantly the bottom of the two minima populated in the atomistic simulation. Actually, it delimits rather clearly the lower boundary of the 1BA2 minima on the left. **Note also that this is only the closing pathway in the absence of ligand; the opening or closing transitions in**

the presence of ribose could populate the undersampled area.

-5'-NTase. The reverse Climber and eBDIMS transitions followed a path not observed in MD.

Again, the transition here is not complete and MD only shows the sampling from the unbound end (1HPU) and intermediate (1O18) states; **still, MD populates half of the area delimited by the forward/reverse paths.** Running the simulation in the presence of the ligand (a nucleotide) in the “closing” direction could extend the sampling to the undersampled area.

-RNaseIII. The closest conformation of eBDIMS to the catalytic, functional state is around 10 Å.

As discussed in the text, the catalytic state requires the presence of Mg²⁺ (and the RNA ligand), so it is driven electrostatically. Still, **it is clear that there is a clear deviation in the reverse path approaching the area of the Mg-bound structures, and that is seen visually in the PC1-2 space and mathematically as an inflection point in both rMSD and PC1-2 distance profiles (Fig.4D).**

-All the method seems to be capable to sample the GLIC and SERCA transitions, but will be interesting to see the rMSDs differences.

We provide rMSD as suggested (see above), but as as can be seen, **rMSDs are very similar for all methods and do not relate with the “stability” of the paths projected onto the experimental PC1-2 subspace.** Note also that **crystallographic intermediates are not the true minima in MD, so the isolated rMSD to these intermediates is not an ideal metric to rate the different approaches.** We have added a brief discussion regarding methods performance in the Supplementary material.